# Correlation between subsurface salinity anomalies in the Bay of Bengal and the Indian Ocean Dipole and governing mechanisms

Zheen Zhang[1], Thomas Pohlmann[1], and Xueen Chen[2]

[1]Institute of Oceanography, Centre for Marine and Climate Research, University of Hamburg, Hamburg, Germany
[2]College of Oceanic and Atmospheric Sciences, Ocean University of China, Qingdao, China

**Correspondence:** Zheen Zhang (zheen.zhang@uni-hamburg.de)

**Abstract.** Lead-lag correlations between the subsurface temperature/salinity anomalies in the Bay of Bengal (BoB) and the Indian Ocean Dipole (IOD) are revealed in model results, ocean synthesis, and observations. Mechanisms for such correlations are further investigated using the Hamburg Shelf Ocean Model (HAMSOM), mainly on the salinity variability. It is found that the subsurface salinity anomaly of the BoB positively correlates to the IOD with a lag of three months on average, while the subsurface temperature anomaly negatively correlates. The model results suggest the remote forcing from the equatorial Indian Ocean dominates the interannual subsurface salinity variability in the BoB. The coastal Kelvin waves carry signals of positive (negative) salinity anomalies from the eastern equatorial Indian Ocean and propagate counterclockwise along the coasts of the BoB during positive (negative) IOD events. Subsequently westward Rossby waves propagate these signals to the basin at a relatively slow speed, which causes a considerable delay of the subsurface salinity anomalies in the correlation. By analyzing the salinity budget of the BoB, it is found that the diffusion dominates the salinity changes near the surface, while the advection dominates the subsurface; the vertical advection of salinity contributes positively to this correlation, while the horizontal advection contributes negatively. These results suggest that the IOD plays a crucial role in the interannual subsurface salinity variability in the BoB.

## 1 Introduction

The Bay of Bengal (BoB) is a monsoon-controlled tropical ocean located in the northeast of the Indian Ocean. The robust monsoon significantly influences the ocean circulation, vertical water exchange, and water characteristics in the BoB (Shetye et al., 1991, 1996; Vecchi and Harrison, 2002; Li et al., 2017). During the summer monsoon, the Southwest Monsoon Current brings saltier Arabian Sea water into the BoB, whereas the Northeast Monsoon Current brings fresher water from the BoB to the Arabian Sea during the winter monsoon (Vinayachandran et al., 1999; Jensen, 2001; Sanchez-Franks et al., 2019). Water from the Arabian Sea also enters the BoB as a subsurface flow during the northeast monsoon, which is proved by observations and model works (Wijesekera et al., 2015; Gordon et al., 2016). The salinity exchanges between the BoB and the equatorial Indian Ocean also show a seasonality associated with the monsoon (Jensen et al., 2016; Trott et al., 2019). In addition to the local monsoon, remote forcing from the equator also affects the ocean circulation and thermocline in the BoB, by which equatorial signals pass through the Andaman Sea (Potemra et al., 1991; Yu et al., 1991; McCreary et al., 1993, 1996;

Girishkumar et al., 2013). The Andaman and Nicobar Islands, as well as the eastern border of the Andaman Sea, significantly alter the circulation in the BoB (Chatterjee et al., 2017). A recent numerical study suggests that the equatorial forcing plays a dominant role in interannual variations of sea surface height and thermocline in the BoB during Indian Ocean Dipole (IOD) and El Nino/Southern Oscillation (ENSO) events, especially for their spatial pattern (Pramanik et al., 2019).

The IOD is an east-west dipole mode that dominates the interannual sea surface temperature (SST) variability in the tropical

Indian Ocean (Saji et al., 1999; Webster et al., 1999; Schott et al., 2009; Deser et al., 2010), and it is physical entity independent of the ENSO (Ashok et al., 2003; Fischer et al., 2005). The spatiotemporal coupling among ocean dynamics, SST, winds, rainfall revealed by the IOD have inspired many studies regarding the relationship and processes between the IOD and variations of surface/subsurface temperature/salinity in the tropical Indian Ocean (Rao et al., 2002; Shinoda et al., 2004; Thompson et al., 2006; Grunseich et al., 2011; Du et al., 2012; Zhang et al., 2013; Sayantani and Gnanaseelan, 2015; Kido and Tozuka, 2017;

Kido et al., 2019a). The research on both sea level and annual mean subsurface temperature anomalies revealed a see-saw of the thermocline that related to the IOD (Saji et al., 1999). During the positive IOD (pIOD) phase, westerly winds weaken, allowing cold water to rise in the eastern equatorial Indian Ocean and warm water to move toward to the west, therefore lifts the equatorial thermocline in the east; during the negative IOD (nIOD) phase, vice versa, and lifts the equatorial thermocline in the west.

Recently it has been reported that the Wyrtki Jets (Wyrtki, 1973) affect the salinity balance of the BoB by means of their northward bifurcation (Wang, 2017). Furthermore, it could be shown, that on the interannual time scale, the Wyrtki Jets are highly associated to the IOD. In particular, this hold true for the fall jet, which develops in the peak phase of the IOD (Nyadjro and McPhaden, 2014; McPhaden et al., 2015). Previous studies have discussed the impact of IOD on the subsurface dynamics in the equatorial Indian Ocean. However, mechanisms and quantitative understandings for the impact of IOD on the subsurface

dynamics in the BoB have not been established yet, especially for the impact on subsurface salinity. Subsurface salinity is of great importance for determining the ocean barrier layer and mixed layer depth (Lukas and Lindstrom, 1991; Montégut et al., 2007; Li et al., 2018; Kido et al., 2019b). Understanding the variations and dynamics of the subsurface salinity is helpful to understand the evolution of stratification and upper ocean properties, to further understand the response of the ocean to the atmosphere and its role on climate. Nevertheless, it is not clear how the subsurface salinity in the BoB varies and whether it

is affected by the IOD. The questions addressed here are; is there an identifiable correlation between the subsurface salinity in the BoB and the surface temperature in the tropical Indian Ocean on the interannual scale, how are these two variabilities related, how does the subsurface salinity in the BoB respond to the IOD.

To answer the above questions, the subsurface salinity variability in the BoB and its relation with the IOD and the corresponding mechanisms are investigated in this paper. Unless otherwise specified, anomalies used in this paper are residuals

subtracting monthly climatology from monthly data. The rest of this paper is organized as follows. In section 2, we introduce four data sets and a regional ocean model used in this study, and the model validation is also presented in this section. In section 3, we examine the correlation between the subsurface temperature/salinity anomaly of the BoB and the IOD through analyzing the four independent data sets and the model results. Connecting mechanisms and contributions of advection and diffusion are discussed in section 4. Section 5 gives the summary and discussion.

**Table 1.** Summary of Data Sets

| Data set | Grid [°] | Period | Type |
|----------|----------|--------|------|
| EN4 | $1 \times 1$ | 1951-2005 | global quality controlled monthly objective analyses |
| GECCO2 | $1 \times 1$ | 1951-2005 | ocean synthesis |
| MPI-ESM-MR | $0.4 \times 0.4$ | 1951-2005 | free run under historical condition |
| RC_Clim | $1 \times 1$ | 2004-2018 | Argo-based data |

## 2 Data and model

### 2.1 Data sets

In order to examine the potential correlation between the surface temperature pattern in the tropical Indian Ocean and the subsurface salinity variability in the BoB on the interannual scale, four independent data sets (Table 1) are used in this study. The first is the global quality controlled monthly ocean temperature and salinity objective analyses of version 4.2.1 of the Met Office Hadley Centre 'EN' series (Good et al., 2013), named as EN4. The second is an ocean synthesis, which is the German contribution of the Estimating the Circulation and Climate of the Ocean project GECCO2 (Köhl, 2015). The third is the free run of mixed resolution of MPI-ESM (Jungclaus et al., 2013) under historical condition, named as MPI-ESM-MR. The fourth is the Roemmich-Gilson Argo Climatology (Roemmich and Gilson, 2009), named as RG_Clim, which offers a basic description of the modern upper ocean based entirely on Argo data. Due to the limitation of the data period, monthly anomalies of RG_Clim are defined on its monthly climatology from 2004 to 2016. Monthly anomalies of the other three data sets are defined on their monthly climatology from 1971 to 2000.

### 2.2 Model setting

For the purpose of discussing the relevant processes and mechanisms, a regional ocean model is performed. The Hamburg Shelf Ocean Model (HAMSOM) we applied in this study is a three-dimensional baroclinic primitive equation model based upon a semi-implicit numerical scheme (Backhaus, 1985; Pohlmann, 1996, 2006). In contrast to explicit shelf sea models, the semi-implicit scheme proposed is faster and allows the simulation of the shelf and the deep ocean regions together without being limited by stability considerations for the free surface (Backhaus, 1985). The underlying primitive equations are defined in z-coordinates and Arakawa C-grid under the hydrostatic and Boussinesq assumption. For temperature and salinity, the second order Lax-Wendroff scheme is applied for advection, the horizontal eddy viscosity is defined according to Smagorinsky diffusivity (Smagorinsky, 1963), and vertical viscosity is calculated using the Kochergin scheme (Pohlmann, 1996, 2006).

In principle, we perform a dynamic downscaling simulation on the model domain using HAMSOM with the external forcing derived from MPI-ESM-MR historical scenario. The model domain (Figure 1b) covers the Bay of Bengal and the Andaman Sea, ranging zonally from $77.4°E$ to $103.5°E$ and meridionally from 0 to $22.83°N$, with bathymetry derived from SRTM30_PLUS (Becker et al., 2009). The horizontal model resolution is set to $5' \times 5'$. A total of 58 model layers are specified in vertical, of

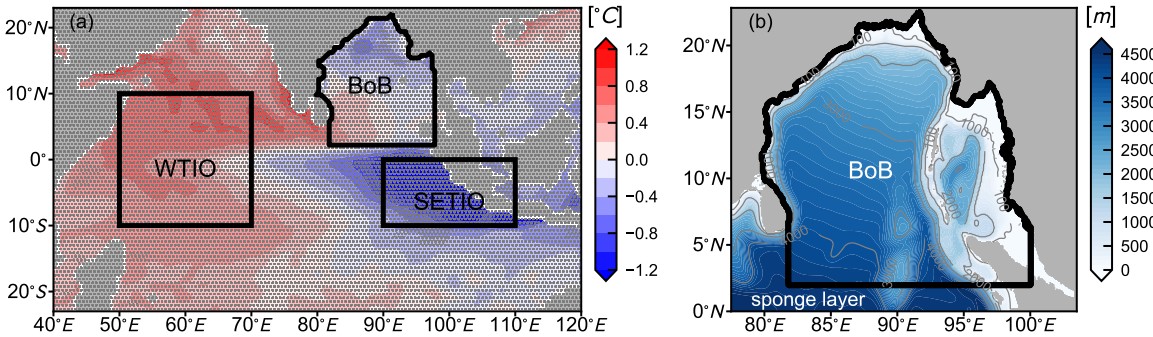

**Figure 1.** Composite of sea surface temperature anomalies during ASO of pIOD years from MPI-ESM-MR (a). The contour intervals are 0.2°$C$. Anomalies significant at the 95% confidence level by a two-tailed Welch's t-test are hatched with grey dots. Western tropical Indian Ocean (WTIO) and southeastern tropical Indian Ocean (SETIO), the two areas related to the dipole mode index (DMI), are marked with black boxes. The bathymetry used in the downscaling simulation (b). Our research domain, the Bay of Bengal (BoB), is marked with black borders in (a) and (b).

which there are 26 layers over upper 200 meters and 33 layers over upper 400 meters. To stabilize the inner model domain, a sponge layer is implemented along the lateral open boundaries in order to damp disturbances arising from inconsistencies within the prescribed boundary conditions extracted from the MPI-ESM-MR. Hence, equatorial processes that are not directly resolved in our model domain are still able to enter the inner domain through the prescription of open boundary conditions. A correlation analysis (not shown) could demonstrate that the sponge layer does not block the propagation of low-frequency

signals (seasonal scale and below). Therefore, only the BoB region (marked in Figure 1b) is analyzed in our HAMSOM simulation.

    Figure 1a offers an overview of our research domain and the tropical Indian Ocean, as well as showing a composite of sea surface temperature anomalies (SSTa) during August-October (ASO, for ease of presentation, the months in the following are simplified to initials) of pIOD years from MPI-ESM-MR. The pIOD years (1974, 1978, 1993, 1997, 2000) are identified by the

normalized dipole mode index (DMI) calculated from the data set itself, in the places with peaks above two and located around September (see Figure 7d). This distribution of composited SSTa (Figure 1a) shows a significant dipole mode in the tropical Indian Ocean, which is in good agreement with previous studies (Webster et al., 1999; Deser et al., 2010). The corresponding DMI time series (Figure 7d) exhibits reasonable interannual variation characteristics. These indicate that the global model used in this downscaling study can realistically reproduce IOD events.

Sea level height, temperature, and salinity at lateral boundaries are monthly prescribed and derived from the oceanic part MPI-OM of MPI-ESM-MR. Atmospheric forcing, such as air temperature, cloud cover, precipitation, specific humidity, air pressure, wind stress, and wind speed at the open boundary, are six-hourly prescribed and derived from the atmospheric part ECHAM6 of MPI-ESM-MR. Under the consideration of a large amount of freshwater input through river discharge to our research domain, the river discharge is also prescribed six-hourly derived from ECHAM6. We applied a bias correction

for forcing parameters on the climatological scale in order to bring the climatology of our simulation closer to the reality, because they are extracted from a purely free run. Principally, the monthly climatology of the external forcing was corrected by reference data by this bias correction procedure. The reference data for atmospheric forcing except air pressure are extracted from ERA5 (Hersbach et al., 2020). The air pressure kept unchanged since it only affects sea surface height due to the inverse barometer effect in HAMSOM, and its seasonal pattern matches well with the local monsoon system. The reference data for

sea temperature and salinity are derived from the World Ocean Atlas 2018. The amplitude of river discharge was corrected by WaterGAP (Döll et al., 2003), and the location where the river discharge enters the ocean was also corrected. Although we applied the bias correction, the interannual signals from MPI-ESM-MR have not been changed, so the IOD signal input to our regional model is consistent with MPI-ESM-MR. Nevertheless, it is noteworthy that HAMSOM is a regional uncoupled ocean model, so no response of the atmosphere to the ocean is considered. Therefore, our model result must be treated as a pure

response of the ocean to external signals rather than a two-way air-sea coupling simulation.

     The simulation runs from 1951 to 2005 with 3 minutes time step and daily average output. In addition to typical output like temperature, salinity, and velocity, six terms concerning salinity change rate related to the contribution of advection and diffusion at U, V, W directions are conducted separately. Terms estimated from monthly outputs may differ significantly from those directly outputted by an online calculation, especially when high-frequency changes occur (Hasson et al., 2013; Köhler

et al., 2018). This so-called 'online analysis' avoids the problem of large residual when directly using monthly data and allows us to precisely close the salinity budget and track relevant exchange processes of salinity.

## 2.3    Model Validation

Before investigating the interannual variability and detailed mechanisms of subsurface salinity in the BoB, the HAMSOM result is validated on the climatological scale by comparing it with other data sets. Figure 2 shows their spatial pattern of

climatological surface and subsurface salinity. River discharge and distribution affect this spatial pattern at the surface, as well as the saline water from the western boundary. Reasons for the subsurface salinity pattern are complicated, ocean circulation and upwelling/downwelling systems may be involved. Both in the surface and in the subsurface, the climatological salinity from HAMSOM presents a gradient from southwest to northeast, which is more consistent with both individual data sets GECCO2 and EN4 than from MPI-ESM-MR. Hence it can be concluded that the bias correction we applied here improves our

modeling by offering a more realistic climatological background.

     Noteworthy that the seasonality is one of the most crucial characteristics in the research region. For the overall monthly climatology of salinity, HAMSOM results also show a reliable seasonal variability (Figure 3). At the surface, the significant seasonal salinity variability is supposed to be the consequence of freshwater flux variability caused by the monsoon. All five data sets show a consistent seasonality of the surface, which indicates the domination of monsoon in this region. The

monthly climatological salinity of these data sets differs more for the subsurface than for the surface. The averaged Pearson correlation between each line shown in Figure 3b is 0.63, 0.42, 0.60, 0.73, and 0.68 for EN4, GECCO2, RG_Clim, MPI-ESM-MR, and HAMSOM, respectively. The lack of subsurface observations and more complex subsurface thermodynamics and hydrodynamics can be the reasons for the difference.

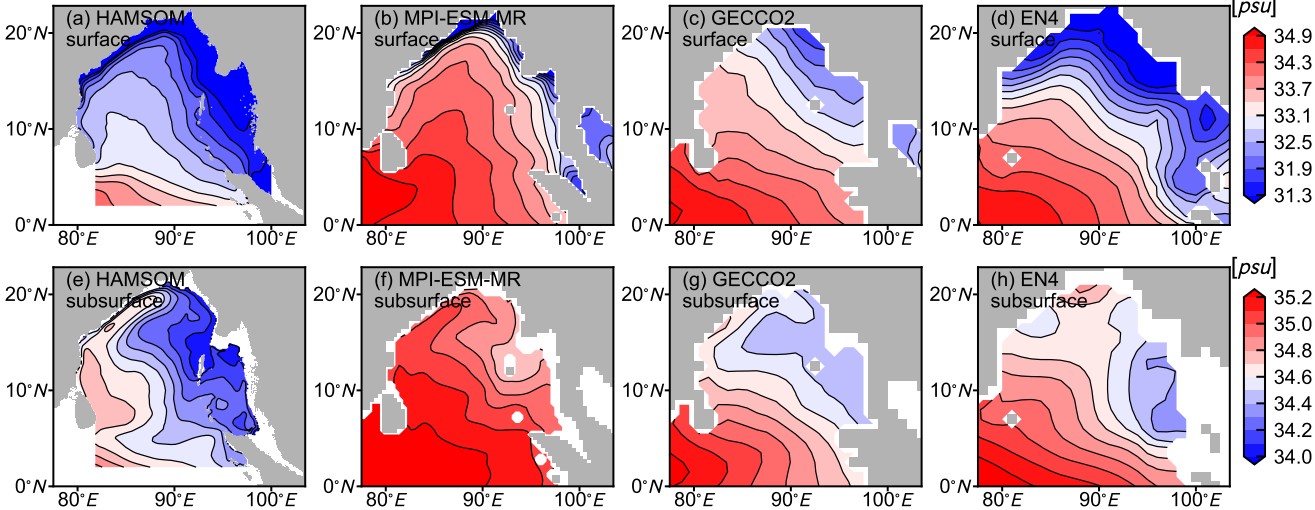

**Figure 2.** Spatial pattern of climatological surface (a, b, c, d) and subsurface (100 $m$; e, f, g, h) salinity from HAMSOM, MPI-ESM-MR, GECCO2, and EN4, respectively. The contour intervals are 0.3 $psu$ for surface, but 0.1 $psu$ for subsurface.

The upper ocean circulation is also validated in two sections (Figure 4 and Figure 5). In general, circulations from HAMSOM is in well agreement with those from MPI-ESM-MR and GECCO2. The direction of upper ocean currents is reversed in MJJ and NDJ, which indicates that the monsoon dominates the upper ocean flow field in the BoB. Given the higher model resolution and more accurate terrain, HAMSOM is expected to perform better in coastal areas. The western boundary current simulated by HAMSOM, also known as the East Indian Current, is stronger than that given by GECCO2 (Figure 4), which should be attributed to the higher resolution.

Figure 6 shows the Taylor diagram (Taylor, 2001) of the surface and subsurface salinity from HAMSOM and other data sets. In this Taylor diagram, the standard deviation reflects both the temporal variability and the spatial variability. The surface salinity standard deviation of HAMSOM is consistent with the observation-based EN4, indicating the realistic extent of HAM-SOM in simulating realistically the amplitude of variations. Even though HAMSOM has a finer grid than EN4, the sea surface feature simulated by HAMSOM is largely determined by the coarser atmospheric forcing due to our simulation strategy, so a good agreement of surface salinity variability between HAMSOM and the reference data set is expected. For the subsurface salinity, the standard deviation of HAMSOM is larger than EN4. Considering that the high horizontal resolution of HAMSOM data allows more mesoscale features, while the low resolution of other data sets does not, the relatively large standard deviation of HAMSOM subsurface salinity is acceptable since it shows more spatial variabilities. The RMS difference is often used to quantify differences in two fields. Comparing to GECCO2 and MPI-ESM-MR, HAMSOM shows a relatively large difference to the reference data set EN4, the resolution difference can be the reason. Overall, HAMSOM simulated salinity variabilities are in a reasonable range.

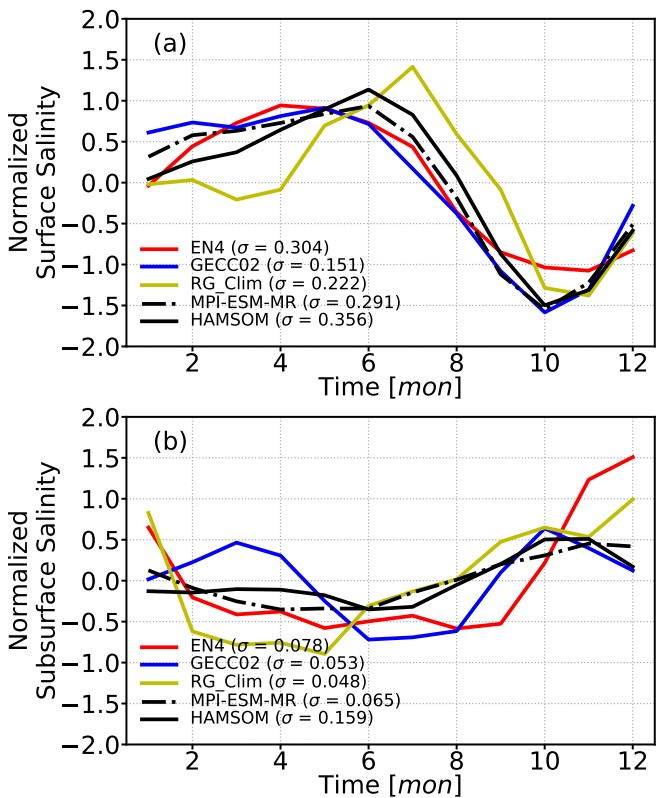

**Figure 3.** Normalized monthly climatology of surface (a) and subsurface (b) salinity of the BoB from EN4, GECCO2, RG_Clim, MPI-ESM-MR, and HAMSOM, respectively. The standard deviation $\sigma$ corresponding to each data set is labelled.

The above validation indicates that HAMSOM model can reproduce reasonable climatological fields and is reliable to be used as a numerical approach to study the physical processes and their specific contributions for the BoB. The interannual variations simulated by HAMSOM are combinations of external signals from MPI-ESM-MR and internal variabilities produced by HAMSOM itself. Hence, it can be concluded that it is reasonable to discuss the interannual variability and corresponding physical processes simulated by HAMSOM in the following sections.

## 3 Lead-lag correlation

Four individual data sets and the downscaling model results are used in this section to examine if there is a statistically significant relationship between the subsurface temperature/salinity anomalies of the BoB and the SSTa of the tropical Indian Ocean on the interannual scale, specifically, the IOD. The DMI describes the difference in SSTa between the western tropical Indian Ocean (WTIO) and the southeastern tropical Indian Ocean (SETIO, see Figure 1). It has a strong correlation with the principal component of EOF2 in the tropical Indian Ocean and is considered to be a reliable representation of the IOD (Saji

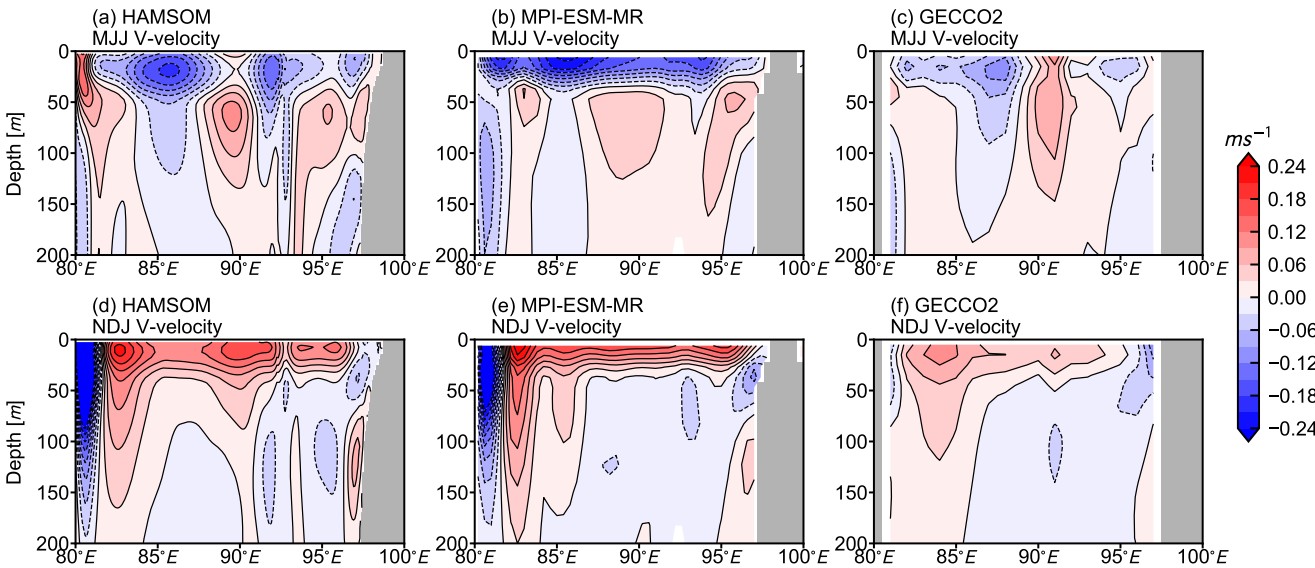

**Figure 4.** Depth-longitude section of climatological V-velocity (averaged over $10°N$ to $12°N$) during MJJ (a, b, c) and NDJ (d, e, f) from HAMSOM, MPI-ESM-MR, and GECCO2, respectively. The contour intervals are $0.03\ ms^{-1}$.

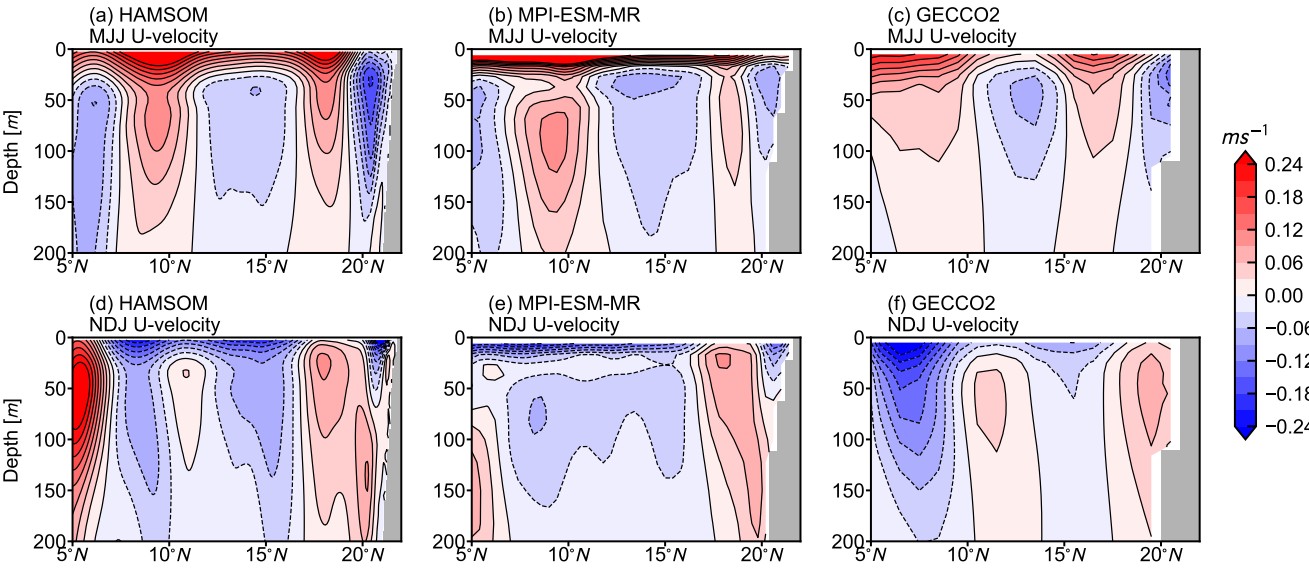

**Figure 5.** Depth-latitude section of climatological U-velocity (averaged over $88°E$ to $90°E$) during MJJ (a, b, c) and NDJ (d, e, f) from HAMSOM, MPI-ESM-MR, and GECCO2, respectively. The contour intervals are $0.03\ ms^{-1}$.

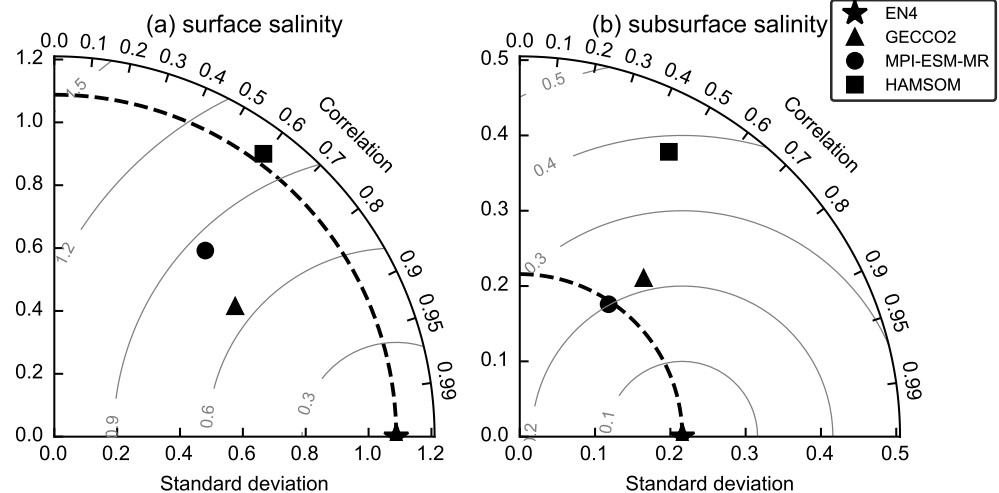

**Figure 6.** The Taylor diagram of (a) the surface salinity and (b) the subsurface salinity from 1971 to 2000 for different data sets. The observation-based EN4 (pentagon) is chosen as the reference data set. Grey lines indicate the centered RMS difference from the reference data set.

et al., 1999). The time series of DMI can indicate different phases of the IOD, so in this study, DMI also covers the meaning of IOD variability.

In order to focus on interannual variations, a 3-month running mean is applied on the monthly time series of DMI and the domain averaged subsurface temperature and salinity anomalies of the BoB. Normalized time series from HAMSOM, MPI-ESM-MR, GECCO2, and EN4 are shown in the first column of Figure 7. Subsurface is defined at a depth of 100 $m$, where wind-induced mixing is negligible, while upwelling/downwelling still plays a role. The DMI time series of HAMSOM is extracted from MPI-ESM-MR. DMI extracted from different data set have similar interannual variation characteristics,

but they do not precisely match. EN4 and GECCO2 both well capture some typical pIOD events like in the year 1994 and 1997. While the free run of MPI-ESM-MR shows a reasonable amplitude and interannual variations and does not exactly repeat the positive event in 1994, which has to be expected, since in this historical run only the statistical features have to be consistent. Lead-lag running Pearson correlation coefficients with a window of 30 years between the DMI and the domain averaged subsurface temperature/salinity anomaly of the BoB (as indicated in Figure 1b) are calculated. The value of the

Pearson correlation determines the extent of linearity between two variables. All these four data sets show that the subsurface temperature anomaly of the BoB negatively correlates to the DMI with a notable lag of about three months on average. Results from HAMSOM, MPI-EMS-MR, and GECCO2 also show a similar but positive correlation between the subsurface salinity anomaly of the BoB and the DMI, but results from EN4 do not.

At the same time, by comparing the correlation magnitudes obtained from different data sets, it is noticeable that the corre-

lation is stronger when the data set shows a lower degree of freedom. The term "degree of freedom" is used here to describe the inherent complexity of a data set, and this complexity is mainly determined by the number of processes involved in the data

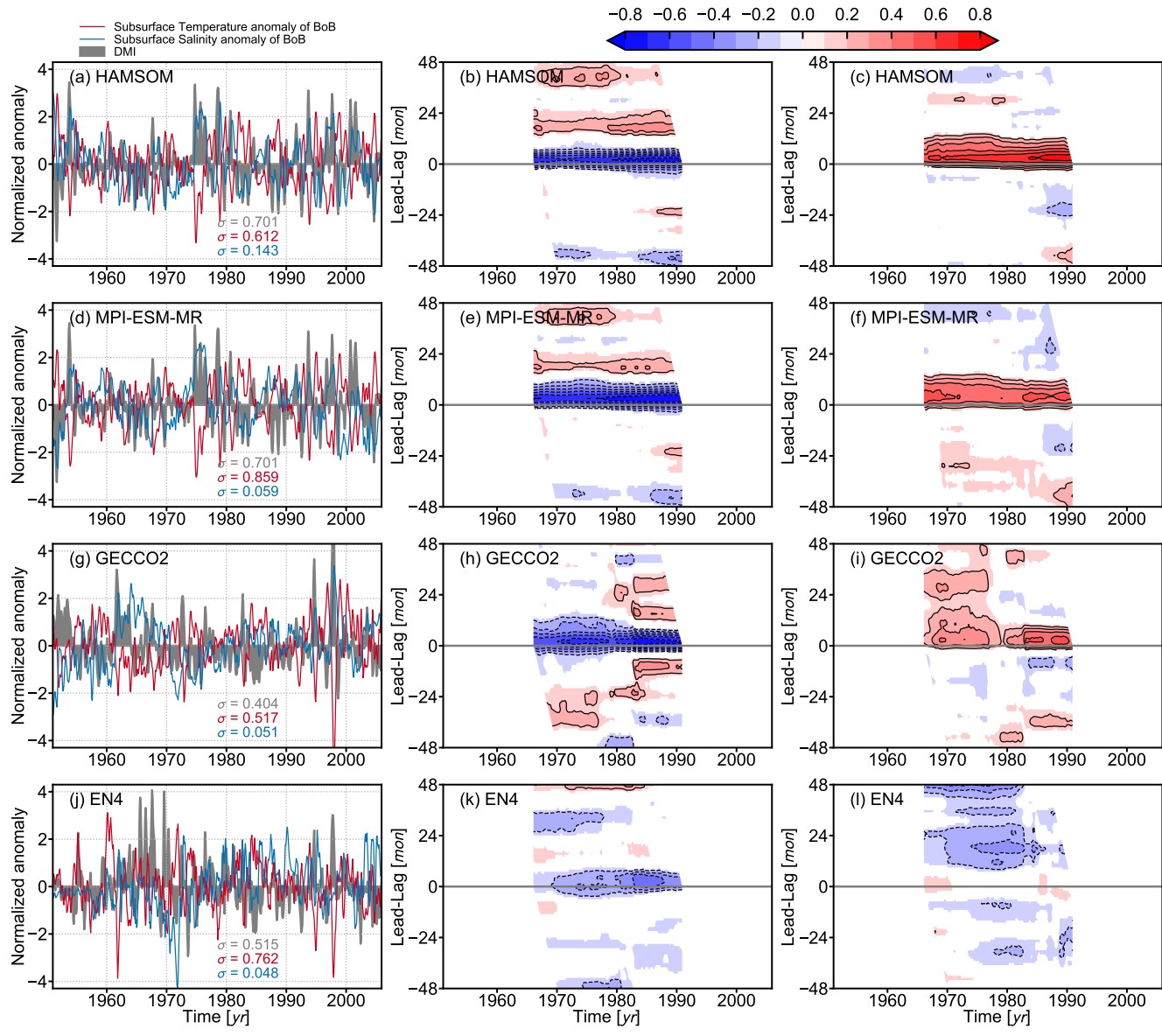

**Figure 7.** Normalized 3-month running mean of DMI, temperature anomaly and salinity anomaly at subsurface (100 $m$) of the BoB from HAMSOM (a), MPI-ESM-MR (d), GECCO2 (g), and EN4 (j), respectively, are shown in the first column. The standard deviation $\sigma$ is labelled with the corresponding color. Lead-lag running Pearson correlation coefficient with a window of 30 years between the DMI and the subsurface temperature (salinity) anomaly from each data set is shown in the second (third) column, respectively. The contour intervals are 0.1. Only significant correlation coefficients with $p-value < 0.05$ are shaded.

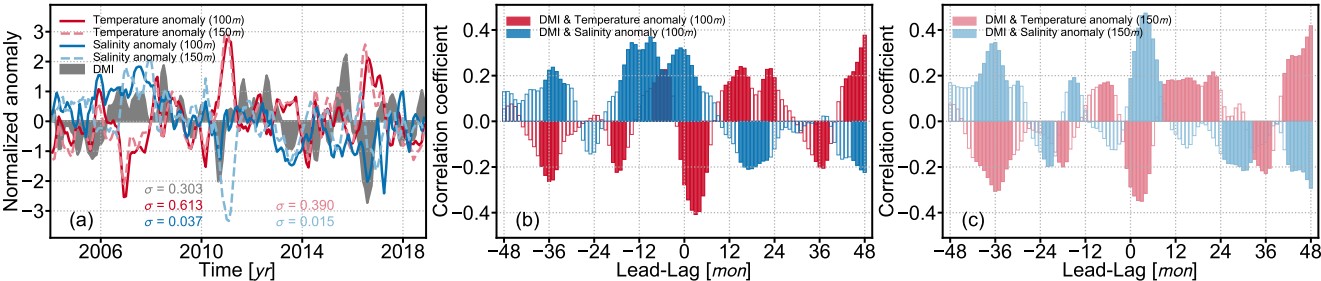

**Figure 8.** Normalized 3-month running mean of DMI, temperature anomaly and salinity anomaly at subsurface of the BoB from Argo based RG_Clim are shown in (a). The standard deviation $\sigma$ is labelled with the corresponding color. Their respective lead-lag relations described by Pearson correlation coefficient between the DMI and subsurface anomalies are shown in (b, of $100\ m$) and (c, of $150\ m$). Only significant correlation coefficients with $p-value < 0.05$ are shaded.

set itself. For example, HAMSOM has a lower degree of freedom than MPI-EMS-MR because it is a regional ocean model not including the ocean-atmosphere feedback processes. GECCO2 has a higher degree of freedom because assimilation processes are included. EN4 is supposed to have the highest degree of freedom of these four data sets because it is based on observations.

The difference in correlation magnitudes between them can be explained by the difference in their own degrees of freedom. Therefore, it can be reasonably inferred that this correlation does exist in the real ocean, but at the same time, there are some processes in reality that obscure the correlation. In this sense, our HAMSOM simulation is more suitable to investigate the physical processes behind the presented correlation.

The lack of observations and the objective analysis method used in EN4 limits its capability to reproduce the interannual 195 subsurface salinity variability in the BoB. There is a weight index from 0 to 1 in EN4 which states the total weighting given to the observation increments when forming this analyses, and the mean weight of subsurface temperature and salinity for the BoB is 0.57 and 0.22, respectively, which points out the lack of salinity observations in the BoB. For example, there are almost no observations from 1951 to 1956 for subsurface salinity of the BoB, so the subsurface salinity anomaly shows artificial oscillations during this period (Figure 7j).

The situation of lack of subsurface salinity observations in the BoB has improved since 2000, especially with the development of Argo. We present related time series and their lead-lag relation calculated from RG_Clim in Figure 8. These time series support the correlations described by the other four data sets, and the subsurface salinity anomaly positively correlates with the DMI. The DMI leads the subsurface temperature anomaly, which can be seen for $100\ m$ and $150\ m$ depth. Such a leading relationship corresponding to the subsurface salinity anomaly suggested by model-related results does not show for

$100\ m$, but a broad, positive correlation with a peak value close to 0.4 is clear. Moreover, at $150\ m$ depth, the DMI leads the salinity anomaly for four months with a peak correlation of over 0.4. Although the time length of RG_Clim is not as long as the other data sets, this Argo-based data clearly shows that the subsurface salinity anomaly of the BoB is correlated to the zonal gradient of SSTa in the tropical Indian Ocean.

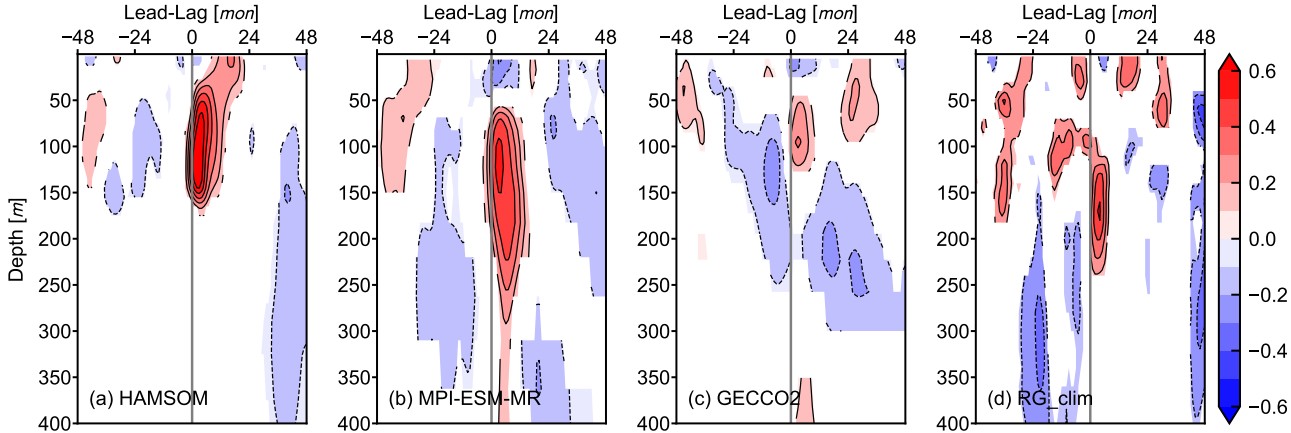

**Figure 9.** Lead-lag Pearson correlation coefficient between the DMI and salinity anomalies of the BoB at different depths from HAMSOM (a), MPI-ESM-MR (b), GECCO2 (c), and RG_Clim (d), respectively. The contour intervals are 0.1. Analysis period is from 1960 to 2005 for (a), from 1951 to 2005 for (b) and (c), and from 2004 to 2018 for (d), respectively. Only significant correlation coefficients with $p-value < 0.05$ are shaded.

Lead-lag Pearson correlation coefficients between the DMI and the salinity anomalies of the BoB at different depths are shown in Figure 9. Three aspects shown by these data sets are noteworthy. First, the most significant positive correlation appears below 50 $m$, and it is possible to be as deep as 250 $m$. Second, the DMI is leading a few months. Third, no obvious positive correlation is validated for the sea surface. These results suggest that the subsurface salinity anomalies of the BoB are indeed related to the IOD with a considerable delay. On average, their correlation reaches its maximum at a three-month delay. The local intense wind-induced mixing and other surface factors that are not closely related to the IOD are the reasons for the upper 50 $m$ of the BoB does not reflect this correlation.

By analyzing time series and their Pearson correlation coefficients, a time-delayed of about three months and positive correlation between the subsurface salinity anomaly of the entire BoB and the IOD represented by the DMI is revealed by observations, ocean synthesis, and modeling. A similar but negative correlation is also revealed between the subsurface temperature anomaly of the BoB and the IOD. The correlation between them differs in different data sets and becomes smaller when the data set has a higher degree of freedom; this is because some other variations may obscure the correlation we are focusing on. However, this correlation can still be detected in the observational data, and it is very significant in the model-related data.

## 4  Mechanisms

In the above analysis, we have determined and discussed the lead-lag correlation between the domain averaged subsurface salinity anomaly of the BoB and the IOD. In this section, we mainly study how are these two variabilities connected by analyzing the HAMSOM result. Besides the connecting mechanisms, related physical processes of BoB's responses to IOD events are also a subject of this section. Several reasons may result in changes in salinity anomalies of the BoB, for example,

the salinity redistribution within the BoB or the salinity exchange between the BoB and its surroundings. Whatever the reason is, it will eventually be reflected in the salinity advection and diffusion. In this manner, an online analysis of the salinity budget is used in this section.

## 4.1 Connecting mechanism

To figure out the general feature of the response of the subsurface salinity in the BoB to the IOD, we construct composites of subsurface salinity anomalies during ASO, NDJ, FMA, and MJJ, of pIOD and nIOD years, respectively (Figure 10). The nIOD years (1979, 1988, 1992, 1998, 2004) are defined similarly to pIOD years, but with valleys below minus two normalized anomaly(see Figure 7d). The pattern of subsurface salinity anomalies is opposite during pIOD and nIOD events in ASO, as well as in NDJ when IOD events end with the DMI is returning to 0. This opposite feature is getting weaker over time, which can be seen in FMA and MJJ. When a pIOD or nIOD event happened in the tropical Indian Ocean, areas near the BoB coasts first show large and statistically significant anomalies (Figure 10a, e). Next, these large and statistically significant anomalies show in most areas of the eastern basin but are limited to the western boundary areas (Figure 10b, f). This developing process is consistent with the characteristics of the coastal Kelvin wave and westward Rossby waves. The Welch's t-test is designed for working with small samples. Of cause, more IOD events would strength the power of the test. However, from our results, the five IOD events are sufficient to show the significant differences between IOD years and climatological conditions, and the propagation of these IOD-related waves in a statistical meaning.

We speculate the propagation process is as follows. First, the subsurface disturbance signals in the eastern equatorial Indian Ocean related to the IOD propagate counterclockwise along the BoB coasts in the form of coastal Kelvin waves. The estimated wave phase speed is about 2.65 $ms^{-1}$, so it takes approximately two weeks to propagate from the equator to the northern BoB (Moore and McCreary, 1990; Cheng et al., 2013). These coastal Kelvin waves travel fast explaining why the related significant anomalies first shown-up near the coasts. Subsequently, these signals are reflected at the eastern boundary and propagate westward into the interior of the basin, since the phase speed of Rossby waves is predominantly moving westward. This also explains, why the signal near the western boundary seems to be trapped there. From the significant area of influence in NDJ, these Rossby waves travel slower, which accounts for the domain averaged subsurface salinity anomaly lags the DMI.

Five subareas are selected in order to further investigate the response of different areas to the IOD (Figure 10a, e). Figure 11 shows the typical salinity profiles and the equivalent vertical displacement yielding a 0.6 $psu$ salinity anomaly in the BoB and five subareas. These salinity profiles of subareas indicate the distribution of salinity stratification and their seasonal changes in our research domain. This distribution is consistent with our model validation shown in Figure 2. Due to the influence of river discharge and distribution, as well as the saline water from the open lateral boundary, the salinity stratification shows a weakening gradient from northeast to southwest. As shown in Figure 10, averaged subsurface salinity anomaly of extreme IOD events can reach to about 0.6 $psu$ in areas close to the coast. As indicated in Figure 11, a 0.6 $psu$ salinity anomaly is equivalent to about 20-50 $m$ vertical displacement. Apparently, these displacements are already significant for vertical water motions, and would be larger for some extreme IOD events since the results shown here represent a mean state.

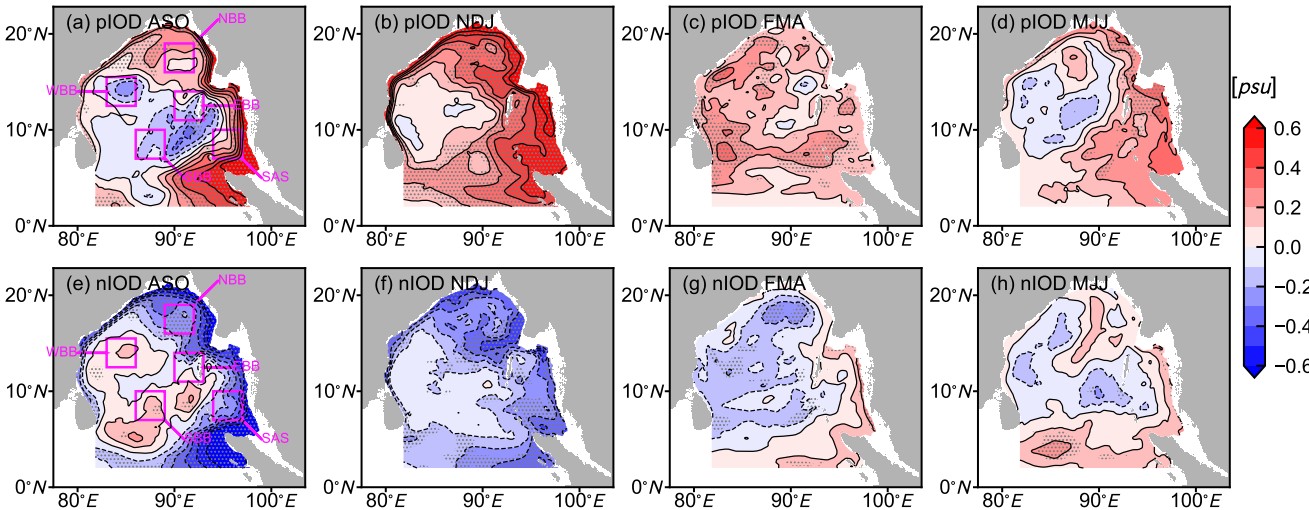

**Figure 10.** Composite of subsurface (100 $m$) salinity anomalies during ASO (a, e), NDJ (b, f), FMA (c, g), and MJJ (d, h), of pIOD and nIOD years, respectively, from HAMSOM. The contour intervals are 0.1 $psu$. Anomalies significant at the 95% confidence level by a two-tailed Welch's t-test are hatched with grey dots. Selected subareas are marked with magenta boxes in (a) and (e).

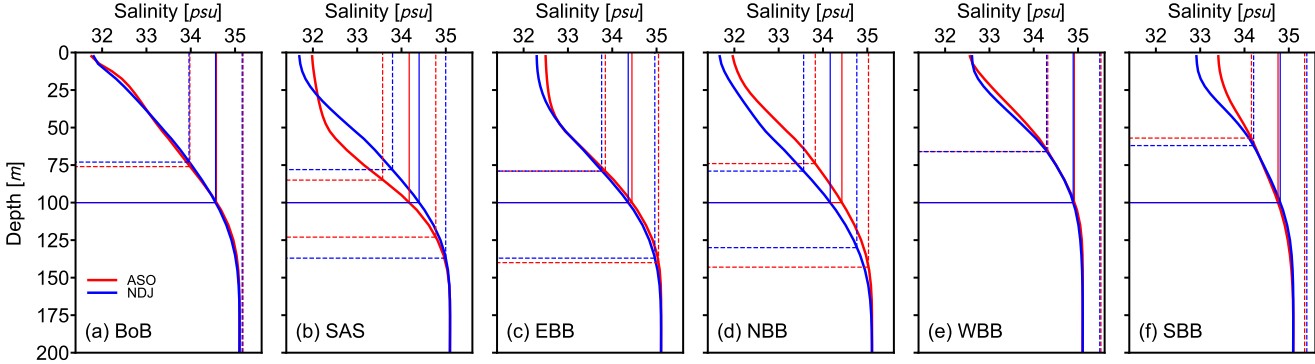

**Figure 11.** Domain-averaged climatological vertical salinity profiles of the BoB (a) and subareas (b, c, d, e, f) during ASO (red thick line) and NDJ (blue thick line). Solid thin lines with corresponding colors indicate the salinity of 100 $m$. dashed thin lines with corresponding colors indicate the equivalent vertical displacement yielding a 0.6 $psu$ salinity anomaly.

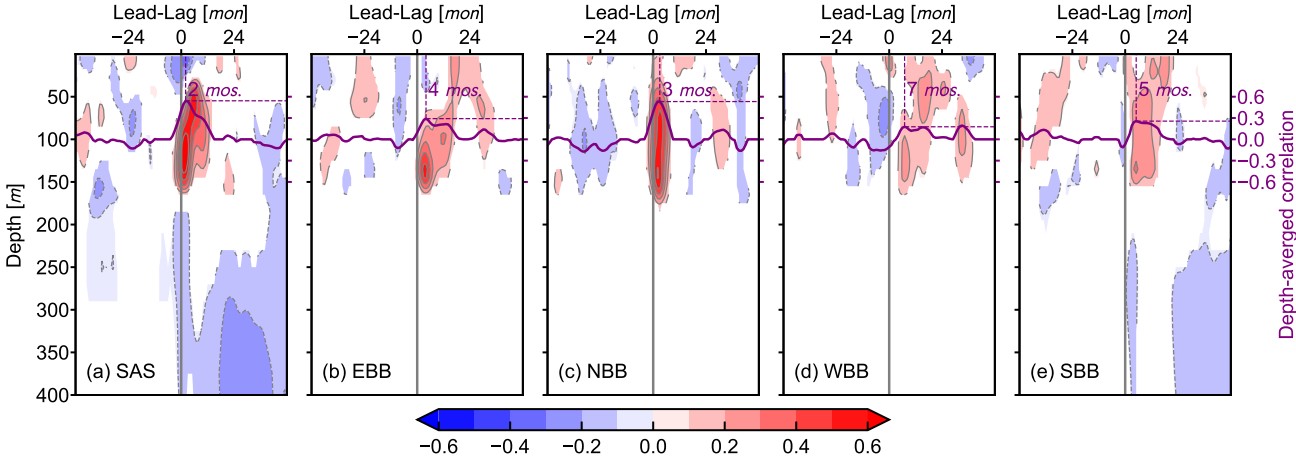

**Figure 12.** Lead-lag Pearson correlation coefficient between the DMI and the salinity anomaly of subareas SAS (a), EBB (b), NBB (c), WBB (d), and SBB (e), respectively, at different depths, from HAMSOM. The contour intervals are 0.1. Analysis period is from 1960 to 2005. Only significant correlation coefficients with $p-value < 0.05$ are shaded. Purple solid lines indicate the depth-averaged correlation over 50-150 $m$. Purple dashed lines indicate the highest correlation and the corresponding lag.

A similar plot as Figure 9 but for subareas is presented in Figure 12. The closer the subarea is to the eastern boundary, the stronger the Pearson correlation between the DMI and the local salinity anomaly. When the subarea is close to but not directly at the western boundary, the correlation is relatively weak, which indicates that the signal is trapped at the west boundary (Figure 12d). Results from the subarea closest to the equator also show weaker correlations (Figure 12e), while results from the subarea that is far away from the equator but closer to the eastern boundary show stronger correlations (Figure 12c), suggesting that the signal propagates along the boundary rather than directly goes north. The lags for these subareas also indicate that the subarea SAS is affected first, then NBB, and then EBB. These features support our speculation that the interannual subsurface salinity variability in the BoB is connecting to the IOD through both coastal Kelvin waves and westward Rossby waves.

It is challenging to observe Rossby waves in the BoB if we only use monthly data because of the basin size. Therefore, daily data from HAMSOM is used for tracking Rossby waves. As the Hovmöller diagram of daily climatological subsurface salinity averaged over $10°N$ to $12°N$ shown in Figure 13a, a westward Rossby wave signal can be seen by the westward low-salinity water. This signal takes approximately four months to cross the basin zonally, and from this, it can be estimated that its propagation speed is about 0.16 $ms^{-1}$. Low-salinity water already appears at the western boundary before the westward Rossby wave had reached here. In May, even though the westward Rossby wave signal represented by the low-salinity water has reached the western boundary, the water at the western boundary is as salty as the water at the eastern boundary, demonstrating that coastal Kelvin waves travel faster and dominate the coastal zone in the BoB.

In pIOD and nIOD years, the propagation characteristics of coastal Kelvin waves and westward Rossby waves are essentially the same as the climatology, but carrying positive and negative anomalies, respectively (Figure 13b, c). These statistically significant anomalies first appear at the eastern boundary, then at the western boundary, then in the basin interior, which

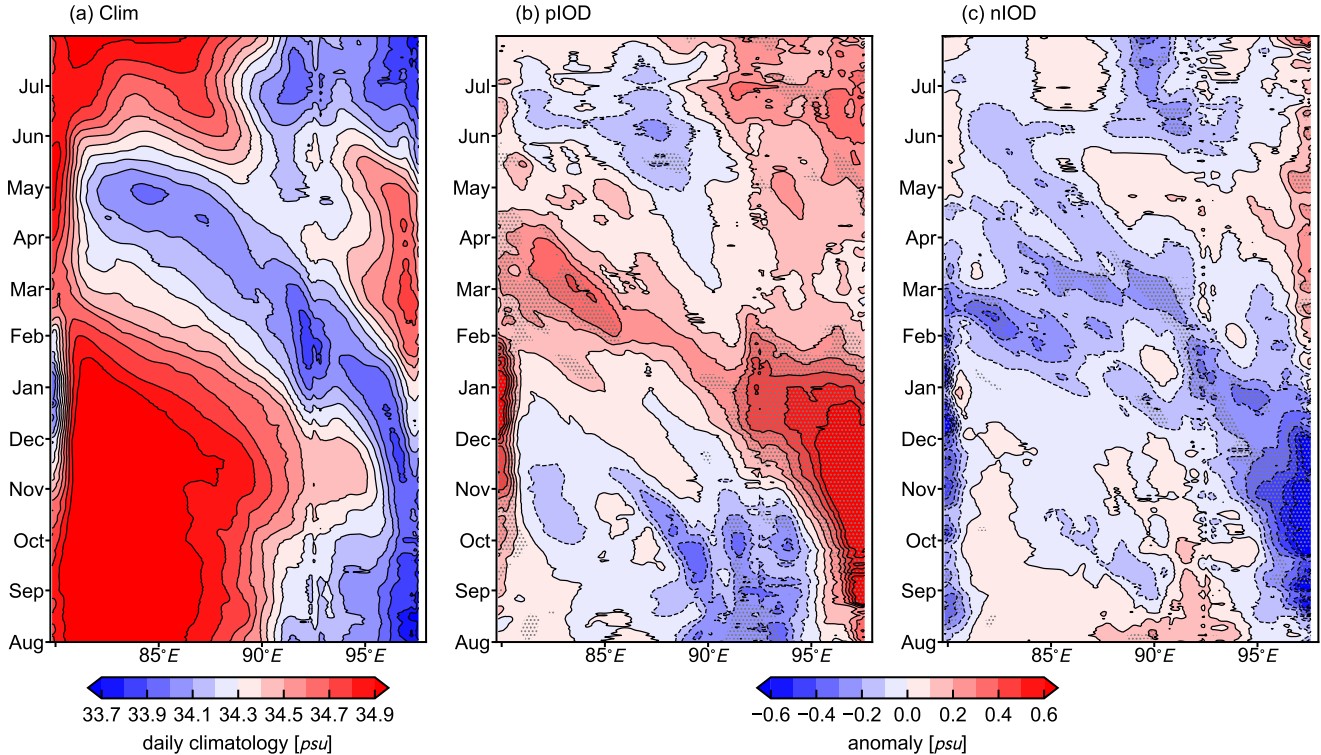

**Figure 13.** Hovmöller diagram of daily climatological subsurface ($100 \, m$) salinity (a; averaged over $10°N$ to $12°N$) from HAMSOM. The intervals are $0.1 \, psu$. (b and c) as in (a), but for composite of subsurface salinity anomalies for pIOD (b) and nIOD (c). The intervals are also $0.1 \, psu$. Anomalies significant at the 95% confidence level by a two-tailed Welch's t-test are hatched with grey dots.

indicates that the extreme IOD signal is propagated to the entire BoB by both coastal Kelvin waves and westward moving
Rossby waves. Previous studies have demonstrated the dominant role of coastal Kelvin waves in sea level variability in the
BoB, especially near the eastern and northern boundaries (Han and Webster, 2002; Cheng et al., 2013). Our analysis about
subsurface salinity anomalies suggests that the coastal Kelvin waves also dominate the western boundary when extreme IOD
events occur. The positive anomalies associated with pIOD reduce the zonal gradients of subsurface salinity, while the negative
anomalies associated with nIOD increase their zonal gradients and result in different baroclinic Rossby wave modes with
different propagating speed.

>   We also calculated the correlation between the local wind and salinity in the BoB and all other subareas (not shown). The
results show that these two parameters are strongly correlated on the seasonal scale, but there is no significant correlation
on the interannual scale. This indicates that interannual signals are primarily not induced by coastal Kelvin waves forced by
the local wind along the east coast of the Andaman Sea, but by far-field signals originating from the equator. Therefore, the
model result suggests that the propagation process through coastal Kelvin waves and westward Rossby waves is the primary
connecting mechanism of the delayed positive correlation between the subsurface salinity anomaly of the BoB and the zonal

SSTa gradient in the tropical Indian Ocean. The interannual variability of thermocline depth in the eastern Indian Ocean is dominated by equatorial Indian Ocean winds, which drive eastward moving equatorial Kelvin waves that are blocked at the Sumatra-Java coasts (Du et al., 2012; Chen et al., 2015). It has been shown that enhanced upwelling occurs in the eastern Indian Ocean during pIOD years (Chen et al., 2016). This enhanced upwelling signal is converted into coastal Kelvin waves, which propagate counterclockwise along the boundary of the BoB. Subsequently, this signal is reflected at the eastern boundary forming westward moving Rossby waves that keep propagating into the central basin. During nIOD events the related subsurface anomalies in the BoB are modulated in a similar way.

## 4.2 Contributions of advection and diffusion

By outputting terms concerning salinity change rate related to the contribution of advection and diffusion, we can precisely close the salinity budget and analyze changes of advection and diffusion in the BoB in different IOD phases. The salinity budget can be written as follows:

$$\frac{\partial S}{\partial t} = -u\frac{\partial S}{\partial x} - v\frac{\partial S}{\partial y} - w\frac{\partial S}{\partial z} + \frac{\partial}{\partial x}\left(\kappa_H \frac{\partial S}{\partial x}\right) + \frac{\partial}{\partial y}\left(\kappa_H \frac{\partial S}{\partial y}\right) + \frac{\partial}{\partial z}\left(\kappa_V \frac{\partial S}{\partial z}\right), \tag{1}$$

where $S$ is salinity, $u$, $v$ and $w$ are zonal, meridional and vertical velocity, respectively, $\kappa_H$ and $\kappa_V$ are horizontal and vertical diffusion coefficients, respectively. The left side represents the salinity tendency (ST), while the right side from left to right represents the salinity change rate of zonal (UADV), meridional (VADV), vertical (WADV) advection, and of zonal (UDIF), meridional (VDIF), vertical (WDIF) diffusion, respectively.

A salinity budget of the BoB at 100 $m$ during ASO is presented in Figure 14. For terms on the right side of the equation 1, at this depth, the advection term is much larger than the diffusion term, and the vertical diffusion term is larger than the horizontal diffusion term. The sum of these large advection terms becomes much smaller and about the same magnitude as the salinity tendency and the vertical diffusion (Figure 14b). All advection terms show significant differences in pIOD and nIOD events comparing to the climatology. Especially during nIOD, the salinity changes caused by advection at each direction increase, which is believed to be the result of increased zonal subsurface salinity gradients associated with nIOD. Meanwhile, it can be seen that at this depth, on the average for the entire BoB, the vertical advection contributes positively to the positive salinity tendency of pIOD and the negative salinity tendency of nIOD. In contrast, the summed horizontal advection contributes negatively.

Figure 15 shows the sum of advection terms, the sum of diffusion terms, and the final salinity tendency at different depths of the BoB and other selected subareas during ASO. The salinity tendency shows a subsurface salinity increase (decrease), indicating a positive (negative) anomaly for pIOD (nIOD) years, which in this season show an obvious response to the IOD signal (Figure 15m, n, o). The results in different regions show that the salinity tendency is dominated by diffusion near the surface, while it is dominated by advection for the subsurface. The dominant role of diffusion near the surface can be explained by the wind-induced mixing. The salinity change rate due to advection shows more obvious responses in the subsurface during both extreme IOD events, especially in the subareas SAS and EBB, and the entire BoB, suggesting that the correlation we are discussing is mainly caused by advection processes.

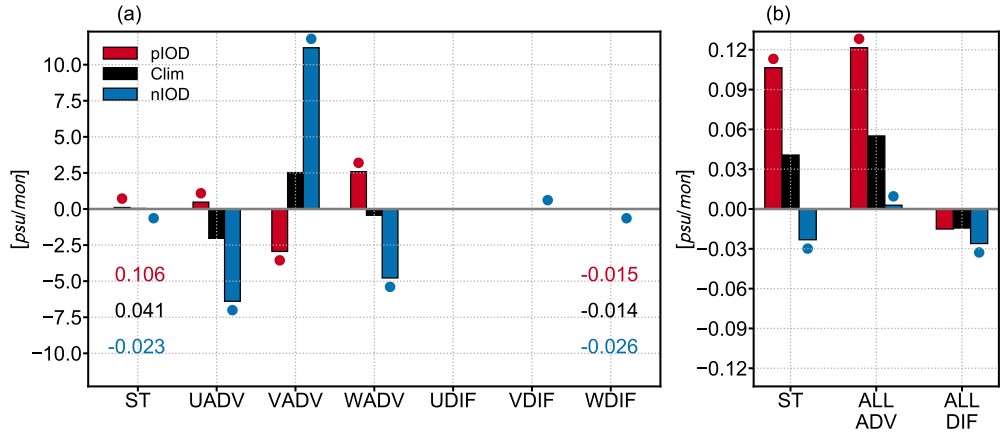

**Figure 14.** Domain-averaged subsurface (100 $m$) salinity tendency and related salinity change rate terms of the BoB during ASO of pIOD years, nIOD years, and climatological period, respectively, from HAMSOM (a). The values of ST and WDIF in different cases are labelled with the corresponding color. The sum of all advection terms and the sum of all diffusion terms are shown in (b). Dots with corresponding color indicate that they are significant different at the 95% confidence level by a two-tailed Welch's t-test comparing to the climatology.

As we analyzed through multiple data sets, there exists a delayed positive correlation between the subsurface salinity anomaly of the BoB and the zonal SSTa gradient in the tropical Indian Ocean represented by the DMI. Therefore, by analyzing the salinity budget of the BoB, the model results suggest that the contribution of advection plays a dominant role in this correlation. Particularly, the vertical advection contributes positively, while the horizontal advection contributes negatively to the correlation stated above.

## 5   Conclusions

In this study, we have investigated the subsurface salinity variability in the BoB on the interannual scale and its relation with the IOD through multiple data sets, and we have also investigated the corresponding mechanisms through a regional ocean model simulation. The regional downscaling model successfully reproduces the reasonable climatology of the salinity and flow field, proving its capability for investigating the physical processes in the BoB. In order to further discuss advection and diffusion contributions to salinity, we have performed an online analysis of salinity budget. This approach can precisely close the salinity budget, and hence, reflects the response of salinity in the BoB to the IOD, also with respect to the driving mechanisms in a quantitative manner.

A delayed positive correlation between the subsurface salinity anomaly of the BoB and the IOD was revealed by analyzing their Pearson correlation coefficient. This correlation is not only shown in the modeling data but also in the ocean synthesis and observations. On average, a lag of three months shows the strongest correlation. Meanwhile, the correlation is relatively weaker when the data set shows a higher degree of freedom, suggesting that some processes exist in reality, which are not well resolved by numerical simulations that may disturb the relation between the subsurface salinity variability of the BoB and the

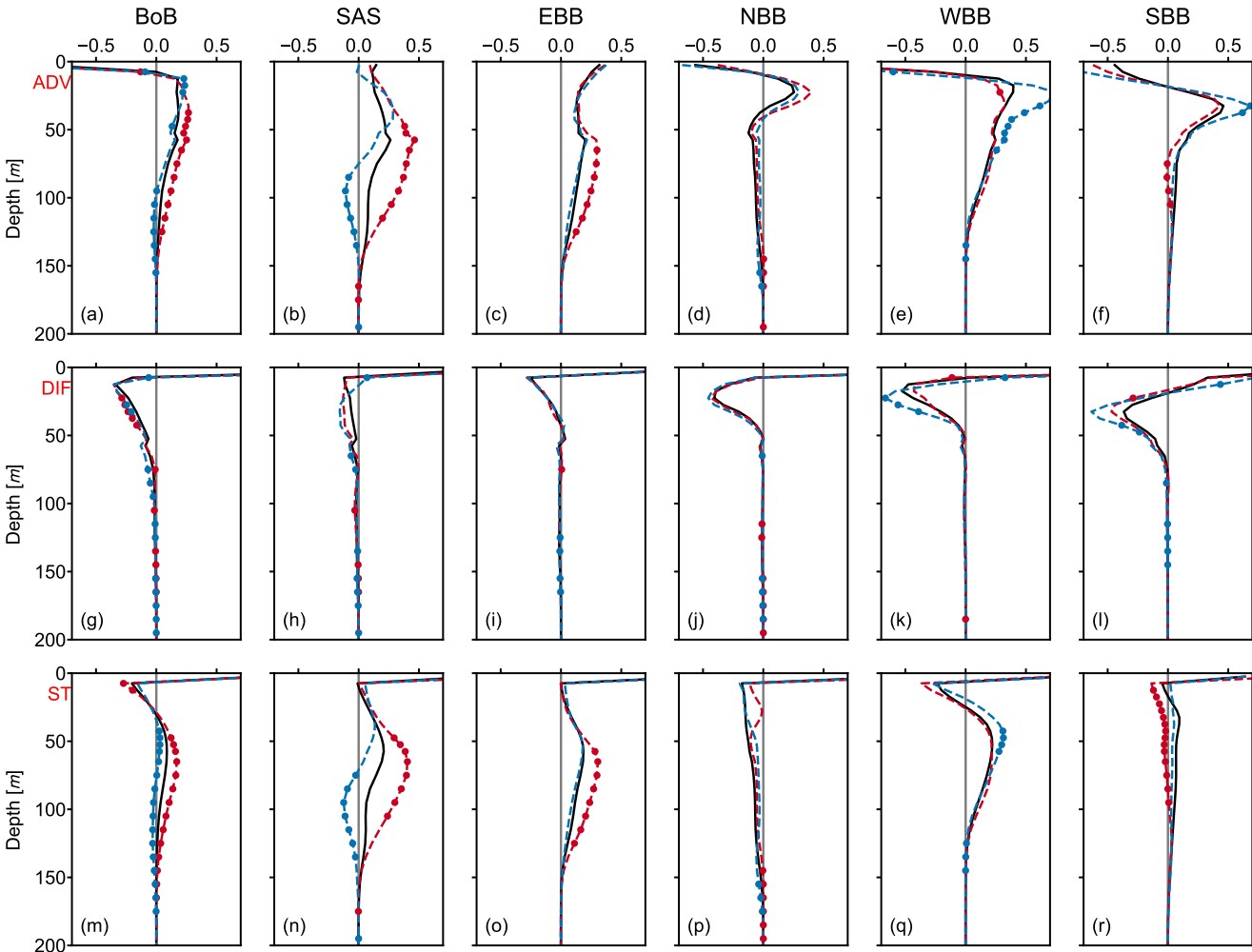

**Figure 15.** Sum of domain-averaged advection terms (in $psu/mon$) at different depths of the BoB (a) and subareas (b, c, d, e, f) during ASO. Black solid line is for the climatology; red and blue dashed line is for the composite of pIOD and nIOD years, respectively. Dots with corresponding color indicate that they are significant different at the 95% confidence level by a two-tailed Welch's t-test comparing to the climatology. The second and third row as in the first row, but for sum of domain-averaged diffusion terms and salinity tendency, respectively.

IOD on the interannual scale. From this perspective, the numerical simulation is a more suitable method for investigating the physical processes behind this correlation.

The model results suggested that the interannual subsurface salinity variability in the BoB and the IOD variability in the tropical Indian Ocean are connected by both coastal Kelvin waves and westward moving Rossby waves. First, coastal Kelvin waves carry the disturbance signal in the eastern equatorial Indian Ocean that is related to the IOD, propagating counterclockwise along the BoB coasts. subsequently, the signal reflects at the eastern boundary and propagates westward to the basin interior in the form of Rossby waves. The main reason that the domain averaged subsurface salinity anomaly lags the DMI several

months is that the westward Rossby waves travel slowly. The analysis of the salinity budget revealed that the contribution of advection plays a dominant role in this correlation. Particularly, the vertical advection shows a positive contribution, while the horizontal advection shows a negative contribution.

    For the eastern equatorial Indian Ocean, the weakening of Wyrtki jet and the strengthening of upwelling caused by the easterly wind anomalies during pIOD result in freshening at the surface and saltening at the subsurface (Kido and Tozuka,

2017). Large-scale wind stress anomalies play the dominant role in the salinity anomalies of this area during IOD events through modulating salinity advection mainly (Kido et al., 2019a). For the BoB, the model results suggest that the remote forcing from the equatorial Indian Ocean converted into coastal Kelvin waves and westward moving Rossby waves is the principal mechanism, which is responsible for the interannual salinity variability in the subsurface. Because of the unique topographic configuration, the BoB is more susceptible to equatorial signals than any other ocean region. Equatorial signals carried by

equatorial Kelvin waves that propagate eastward to west coast of Sumatra, where coastal Kelvin waves are derived, and in turn influence the BoB (Cheng et al., 2013). Correlation analysis shows that the subsurface salinity anomaly positively correlates with the IOD, while the subsurface temperature anomaly negatively correlates, which implies that the IOD remotely modulates the vertical advection in the BoB subsurface. The salinity budget of HAMSOM results proves that the vertical advection positively contributes to the correlation between the subsurface salinity anomaly of the BoB and the IOD. The decomposition

of advective anomalies (Zhang et al., 2013; Li et al., 2016; Kido and Tozuka, 2017) will be helpful to understand the specific contribution of each specific process. For instance, it will allow separately diagnose the contribution of the anomalous vertical salinity gradient and the contribution of the anomalous vertical velocity. During the pIOD phase, intensified upwelling occurs in the eastern Indian Ocean (Nyadjro and McPhaden, 2014; Chen et al., 2016), inducing an uplift of cold, more saline water along the eastern BoB coasts. This anomaly in turn induces coastal Kelvin waves, which are reflected at topographic disturbances,

inducing Rossby waves that move westward to the central basin. Through this chain of processes, the remote forcing from the equatorial Indian Ocean is able to dominate the interannual subsurface temperature/salinity variability in the BoB. The BoB is known as a region with vigorous mesoscale eddy activity (Chen et al., 2012, 2018). How these eddies affect the evolution of subsurface salinity anomalies requires future studies.

    Based on this discovered correlation and related mechanisms, one application is using the DMI to predict the subsurface

ocean state in the BoB. The subsurface ocean state affects the barrier layer and mixed layer depth, as well as the near-surface state and the air-sea energy transfer. However, how the subsurface parameters response to the IOD affects its local upper ocean still needs more studies. Previous studies have demonstrated the importance of forcing from the equator to the BoB, such as

sea surface height, thermocline, and circulation structure (Girishkumar et al., 2013; Chatterjee et al., 2017; Pramanik et al., 2019), especially for the mechanisms forcing the East India Coastal Current (Yu et al., 1991; McCreary et al., 1996; Shankar
et al., 1996). The response of the subsurface salinity field we discussed may also affect the local flow field and mesoscale eddies. Coastal Kelvin waves and westward Rossby waves play a vital role in the process of receiving information from the equator in the BoB, and similarly, they also play their role on the interannual scale. Especially during the nIOD phase, the increase of the zonal subsurface salinity gradients makes it easier to excite high mode westward Rossby waves, suggesting the effect of IOD on the subsurface thermohaline circulation in the BoB. The biological processes are significantly affected by the
salinity stratification and vertical mixing in the BoB (Prasanna Kumar et al., 2002). As our results show, the IOD significantly modulates the BoB subsurface salinity, and further potentially affects the ocean barrier layer and mixed layer depth, through the Kelvin and Rossby waves. Therefore, the correlation and the corresponding processes we discussed are expected to play an important role for the biology of the BoB. For example, the considerable IOD-related vertical displacement may transport nutrients across the halocline and then increase biological production, as the eddy pumping does (Prasanna Kumar et al., 2004).
Furthermore, it is also expected that these waves affect the air-sea exchange processes in the BoB, which in turn influence the remote ocean feedback to the atmosphere.

In addition to these, in the future we will investigate how the relationship between the BoB subsurface parameters and the tropical Indian Ocean surface parameters will be affected by the impact of climate change. The sea surface is more susceptible to global warming, and the BoB subsurface has a notable connection to the tropical Indian Ocean surface. The sea surface
warming may also affect the subsurface and even deeper through dynamic mechanisms. Therefore, how the BoB subsurface responds to climate change is the next subject we are going to study.

*Data availability.* The HAMSOM data are available at https://cera-www.dkrz.de/WDCC/ui/cerasearch/entry?acronym=DKRZ_LTA_119_ds00001. The objective analyses EN4 data are available from this site (https://www.metoffice.gov.uk/hadobs/en4/). The ocean reanalysis GECCO2 data are available in the Integrated Climate Data Center (https://icdc.cen.uni-hamburg.de/). The data of MPI-ESM-MR historical run are available
from the CMIP5 database (https://esgf-node.llnl.gov/projects/cmip5/). The Roemmich-Gilson Argo Climatology data are avaiable from this site (http://sio-argo.ucsd.edu/RG/_Climatology.html). The bathymetric data were obtained from the SRTM30_PLUS (ftp://topex.ucsd.edu/pub/srtm30_plu reference data for bias correction were obtained from ERA5 (https://www.ecmwf.int/en/forecasts/datasets/reanalysis-datasets/era5), WOA18 (https://www.nodc.noaa.gov/OC5/woa18/), and WaterGAP (http://www.watergap.de/).

*Author contributions.* The idea and the methodology was first proposed and discussed by all authors. ZZ deployed the numerical modeling
under the supervision of TP. ZZ performed the data analyse. TP and XC validated the investigation. All authors contributed to the discussion of the results and the review and editing of the manuscript.

*Competing interests.* The authors declare that they have no conflict of interest

*Acknowledgements.* The German Climate Computing Center provided the computing resources. Zheen Zhang was financially supported by the China Scholarship Council. Thomas Pohlmann acknowledges funding from the German Ministry for Education and Research (CLISORM, Grant No. 03F0781A). Xueen Chen acknowledges the National Key Research and Development Plan of China No. 2016YFC1401300.

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
