# Peer review of "Correlation between subsurface salinity anomalies in the Bay of Bengal and the Indian Ocean Dipole and governing mechanisms"

_Ocean Science, 2020_

## Referee Comment (RC1) · Anonymous Referee #1 · 13 Sep 2020

The authors suggested that the subsurface salinity of the Bay of Bengal (BoB) was correlated with the IOD with a lag of several months. They further concluded that the coastal Kelvin waves carry signals of salinity anomalies from the eastern equatorial Indian Ocean and propagate along the coasts. Subsequently, westward Rossby waves propagated these signals to the center basin. I think this study is interesting. However, there are several concerns regarding model validation and result interpretation.

1. Previous studies suggested that remote forcing from the equator significantly modulated the intraseasonal current and eddy kinetic energy (EKE) in the BoB by the coastal Kelvin waves and reflected/free Rossby waves. Strong Intraseasonal Variability of Cur-

rents and large EKE can be found near 5N in the eastern Indian Ocean (e.g., Chen et al. 2017 JPO, 2018 JGR). Obvious salinity anomalies can also be observed here in your Fig. 10.

In Fig.10, you chose several sub-regions. How to choose these sub-regions? To clearly demonstrate your points of remote forcing modulating the salinity by waves, I suggest to choose a sub-region near 5N in the eastern Indian Ocean. Furthermore, it would be helpful to compare the differences of these sub-regions, not just in correlation coefficient (Fig. 11).

2. I am surprised that the amplitude of correlation coefficient for the BoB (Fig. 9) is comparable with that for the sub-regions (Fig. 11). Which region was chosen when calculating the correlation coefficient for the BoB.

3. Could you clearly demonstrate the lagged period for each sub-region? It's not easy to identify this information in Fig. 11.

4. Lines 164-173. Evidences are needed here.

5. Model validation

(1) "The distribution of composited SSTA presents a dipole mode in the tropical Indian Ocean, which indicates the MPI-ESM-MR can reproduce IOD events." It would be better to show more evidences here.

(2) The salinity of HAMSOM is obviously lower than that of the other datasets (Fig. 2). Dose this affect the results in this study?

(3) The salinity in the subsurface layer is less correlated with each other (Fig. 3b). Are we confident enough in the results?

(4) Why is the RC-CLIM not used in Fig. 2? Why choose EN4 as the "standard" in Fig. 6? How to obtain the conclusion "Overall, the standard deviations of HAMSOM and other data sets are in close agreement" according to Fig. 6?

(5) Could you give more explanation of the grey lines in Fig. 6?

---

## Referee Comment (RC2) · Jochen Kämpf (Referee) · 21 Sep 2020

Review of "Correlation between subsurface salinity anomalies in the Bay of Bengal and the Indian Ocean Dipole and governing mechanisms" by Zhang, Pohlmann and Chen.

Overall I enjoyed reading this paper given my own interest in the Indian Ocean Dipole. The underlying approach and analysis seem to be sound and I only have few comments and questions regarding this work. I am uncertain whether the revisions that I call for of minor or major nature.

1) The paper should also start presenting the bathymetry of the regions under discus-

sion.

2) If you discuss salinity anomaly in the Bay of Bengal then you should also present the typical vertical salinity (and temperature) stratification in the region. The salinity anomalies discussed seem to be very, very small compared with vertical gradients in the region. What would be the equivalent vertical displacement yielding the same salinity anomalies?

3) The DMI has a western and an eastern SST signal. What is the influence of the western SST signal on your results? Why didn't you only use the eastern SST anomaly for your correlation analysis? I presume you'd even get a better correlation then. Please test this.

4) It is no surprise at all to see that coastal Kelvin waves and Rossby waves propagate through the oceans. I could think of 1000s of similar studies, just focussing on different regions. To come to the point. Why makes the salinity anomalies in the Bay of Bengal significant? Do these waves play any role in the biology of the regions? Do they create any other climate feedback mechanism? If so, please try to convince the reader of the great significance of your work.

5) So far, only a few pIOD events have been recorded. Does this limited number of events have any implications on the statistics presented in the paper?

6) Why does HAMSOM create such strong NDJ current components of V at 80 degE and U at 5 degN? These are the at the boundaries. Is there any problem with the boundary conditions?

7) If the model domain extends to the equator, please do also show the salinity distributions extending to the equator.
* * *

---

## Author Comment (AC1) · 26 Oct 2020

We thank the reviewer for the careful reading of the manuscript and the constructive comments. Please find below our point-by-point replies:

**Comment 1:**

*Previous studies suggested that remote forcing from the equator significantly modulated the intraseasonal current and eddy kinetic energy (EKE) in the BoB by the coastal Kelvin waves and reflected/free Rossby waves. Strong Intraseasonal Variability of Currents and large EKE can be found near 5N in the eastern Indian Ocean (e.g., Chen et al. 2017 JPO, 2018 JGR). Obvious salinity anomalies can also be observed here in your Fig. 10.*

*In Fig.10, you chose several sub-regions. How to choose these sub-regions? To clearly demonstrate your points of remote forcing modulating the salinity by waves, I suggest to choose a sub-region near 5N in the eastern Indian Ocean. Furthermore, it would be helpful to compare the differences of these sub-regions, not just in correlation coefficient (Fig. 11).*

**Reply 1:**

Yes, a significant intraseasonal signal can also be observed in the southeast of Sri Lanka in our simulation. However, the choice of sub-regions is based on the general circulation pattern in the Bay of Bengal. Essentially, the purpose of these sub-regions is to better describe the propagation of coastal Kelvin waves and westward moving Rossby waves, and to discuss the different contributions of advection and diffusion processes during the wave propagation.

We agree that a sub-region near 5N could help to give us a more comprehensive view, especially with respect to Rossby waves. However, a sub-region at 5N is very close to the southern lateral open boundary, and as shown in Fig. 5d, the U-velocity is overestimated due to open boundary effects, which may result in unrealistic results especially with respect to the salinity advection. Meanwhile, we think the current 5 sub-regions selected, are sufficient for the purpose of our study.

As requested, we will put a stronger focus on the differences of these sub-regions. Therefore, we plan to add a new figure (insert after Fig. 10) showing the vertical salinity profiles in ASO and NDJ, as well as the vertical displacement equivalent to a 0.6 psu salinity anomaly. A new paragraph explaining this new figure will be added.

[Figure]

**Comment 2:**

*I am surprised that the amplitude of correlation coefficient for the BoB (Fig. 9) is comparable with that for the sub-regions (Fig. 11). Which region was chosen when calculating the correlation coefficient for the BoB.*

**Reply 2:**

The region of the BoB is indicated in the Fig. 1, but we admit that this figure is confusing because the BoB is marked on the map from the MPI-ESM-MR. We plan to add a new subplot to Fig. 1

presenting the model bathymetry and our research domain. The exact area, which was used to calculate all BoB-related parameters is also marked in this new Fig. 1b.

[Figure]

**Comment 3:**
*Could you clearly demonstrate the lagged period for each sub-region? It's not easy to identify this information in Fig. 11.*

**Reply 3:**
We plan to replace Fig. 11 by the new figure shown below. The new added purple line give the depth-averaged correlation from 50 m to 150 m. The lagged period for each of the sub-regions is also indicated. Of cause, we will also discuss these correlations and lags in the updated manuscript.

[Figure]

**Comment 4:**
*Lines 164-173. Evidences are needed here.*

**Reply 4:**
We agree. We will rephrase this part making our line of thinking given blow more clear.

The term "degree of freedom" as it is used in this context describes the complexity of a data set (or a system). Results from a general circulation model show a lower degree of freedom than an observational data set because a model is not able to resolve all processes that exist in the real Earth system. Similarly, pure ocean model results show a lower degree of freedom than an atmosphere-ocean coupled model because air-sea feedback processes are not included. Therefore, we think the term "degree of freedom" helps to explain the correlation differences shown in different data sets. And as also mentioned in section 5, lines 302-306, when investigating the principle physical processes behind the correlations we found, the analysis of our HAMSOM simulation results is superior over other data sets, because our model results shows a relatively low degree of freedom.

**Comment 5.1:**

*"The distribution of composited SSTA presents a dipole mode in the tropical Indian Ocean, which indicates the MPI-ESM-MR can reproduce IOD events." It would be better to show more evidences here.*

**Reply 5.1:**

In the updated manuscript we will cite Webster et al., 1999, Deser et al., 2010. These two papers show similar pIOD SSTa pattern. Intended new text: "This distribution of the composited SSTa (Figure 1a) shows a significant dipole mode in the tropical Indian Ocean, which is in good agreement with previous studies (Webster et al., 1999; Deser et al., 2010). The corresponding DMI time series (Figure 7d) exhibits reasonable interannual variation characteristics. These indicate that the global model used in this downscaling study can realistically reproduce IOD events."

**Comment 5.2:**

*The salinity of HAMSOM is obviously lower than that of the other datasets (Fig. 2). Dose this affect the results in this study?*

**Reply 5.2:**

We think this doesn't affect our results very strongly. The reason is that the coastal Kelvin waves and the associated Rossby waves appear as eigen-modes of the stratified ocean. Therefore the characteristics of these waves are mainly determined by the vertical density gradient. And since salinity gradient simulated by HAMSOM are consistent with other third-party datasets, as shown in the model validation section 2.3, we are confident that our final results are reasonable.

**Comment 5.3:**

*The salinity in the subsurface layer is less correlated with each other (Fig. 3b). Are we confident enough in the results?*

**Reply 5.3:**

We calculated the correlation metrics (as shown below) for lines shown in Fig. 3b. From this metrics, we are confident that the seasonal change of the subsurface salinity simulated by HAMSOM is reasonable, which also provides confidence in our final results. We will provide the averaged correlation coefficients (EN4: 0.63; GECCO2: 0.42; RG_Clim: 0.60; MPI-ESM-MR: 0.73; HAMSOM: 0.68) of each datasets in the updated manuscript.

[Figure]

**Comment 5.4:**

*Why is the RC-CLIM not used in Fig. 2? Why choose EN4 as the "standard" in Fig. 6? How to obtain the conclusion "Overall, the standard deviations of HAMSOM and other data sets are in close agreement" according to Fig. 6?*

**Reply 5.4:**
The RG_Clim is an Argo-based dataset, which extends from 2004 to present. The reason why we didn't use RG_Clim in Fig.2 is that the mean state of RG_Clim from 2004 to 2016 presents a different climate state compared to the climate state from 1971 to 2000, especially in light of an increasing global warming.

When plotting a Taylor diagram, the ideal situation would be to have a reference dataset based entirely on observations. Unfortunately, there is no such a dataset available. However, the EN4 dataset shown in Fig. 6 is based on an objective analysis using a global quality control, while other three datasets are model-based. Hence, the EN4 dataset is obviously the best choice, when looking for a reference state.

With regard to the last question, we think the results shown in Fig. 6b are acceptable. To make use of the advantage that HAMSOM results have a high resolution, we employed the HAMSOM grid when calculating the Taylor-diagram-related parameters. Hence, for other datasets, except for HAMSOM, a linear interpolation to the HAMSOM grid was necessary, before performing the Taylor calculation. In particular, for the subsurface salinity, the HAMSOM results are supposed to show more mesoscale features due to HAMSOM's high resolution. In contrast, for the surface salinity simulated by HAMSOM, the spatial patterns are largely determined by the low resolution atmospheric forcing fields. This explains why, compared to the EN4 data, for the sub-surface (Fig. 6b) HAMSOM results show a large standard deviation, whereas for the surface (Fig. 6a) both data sets are in close agreement. We thank the reviewer for pointing out that the explanation of Fig. 6 was not clear enough. We will modify the discussion of Fig. 6 along the lines of the explanation given above.

**Comment 5.5:**
*Could you give more explanation of the grey lines in Fig.* 6?

**Reply 5.5:**
Yes. The grey lines indicate the axis of the centered Root-Mean-Square (RMS) difference, which is often used to quantify differences between two fields. We will improve the explanation of Fig. 6.

---

## Author Comment (AC2) · 26 Oct 2020

We thank the reviewer for the careful reading of the manuscript and the constructive comments. Please find below our point-by-point replies:

**Comment 1:**

The paper should also start presenting the bathymetry of the regions under discussion.

**Reply 1:**

We plan to use the new figure shown below to replace Fig. 1. The new Figure 1b shows the model bathymetry, used in our simulation. The region of the BoB, which we defined for all BoB-related calculations, and the sponge layer are also marked in this new Figure 1b. The sponge layer helps to stabilize the model. We will discuss this latter issue in the updated manuscript.

**Comment 2:**

If you discuss salinity anomaly in the Bay of Bengal then you should also present the typical vertical salinity (and temperature) stratification in the region. The salinity anomalies discussed seem to be very, very small compared with vertical gradients in the region. What would be the equivalent vertical displacement yielding the same salinity anomalies?

**Reply 2:**

Thanks for this very constructive comment, which will improve our manuscript significantly. We plan to add a new figure (given below) showing the vertical salinity stratification in the seasons ASO and NDJ for entire BoB and for the five sub-regions, together with a new paragraph explaining this new figure. Dashed lines indicate the equivalent vertical displacement yielding a 0.6 psu salinity anomaly. Obviously, the vertical salinity gradient close to the surface is relatively large, especially in the northern bay, due to the fact that heavy precipitation and strong river discharge freshens the surface water. This freshening is a local monsoon dominated seasonal feature, which mainly affects the surface and upper ocean. However, at a depth of 100 m, as this new figure indicates, a 0.6 psu salinity anomaly (suggested in Fig.10 & 12) is not negligible. In sub-regions which are significantly affected by coastal Kelvin waves and associated Rossby waves, such as SAS, EBB, and NBB, the equivalent vertical displacement of a positive 0.6 psu anomaly reaches about 20-50 m. By this means, this new figure also gives a clear indication that the wave, which we investigate in our study have a significant influence on the structure of the water column.

---

## Author Response (AR1)

We thank the two reviewers for the careful reading of the manuscript and the constructive comments. Please find below
1, point-by-point replies for Reviewer #1
2, point-by-point replies for Reviewer #2
3, response figures (RF)
4, change-tracking manuscript

**Reviewer #1 Comment 1:**

*Previous studies suggested that remote forcing from the equator significantly modulated the intraseasonal current and eddy kinetic energy (EKE) in the BoB by the coastal Kelvin waves and reflected/free Rossby waves. Strong Intraseasonal Variability of Currents and large EKE can be found near 5N in the eastern Indian Ocean (e.g., Chen et al. 2017 JPO, 2018 JGR). Obvious salinity anomalies can also be observed here in your Fig. 10.*

*In Fig.10, you chose several sub-regions. How to choose these sub-regions? To clearly demonstrate your points of remote forcing modulating the salinity by waves, I suggest to choose a sub-region near 5N in the eastern Indian Ocean. Furthermore, it would be helpful to compare the differences of these sub-regions, not just in correlation coefficient (Fig. 11).*

**Reply 1:**

Yes, a significant intraseasonal signal can also be observed in the southeast of Sri Lanka in our simulation. However, the choice of sub-regions is based on the general circulation pattern in the Bay of Bengal. Essentially, the purpose of these sub-regions is to better describe the propagation of coastal Kelvin waves and westward moving Rossby waves, and to discuss the different contributions of advection and diffusion processes during the wave propagation.

We agree that a sub-region near 5N could help to give us a more comprehensive view, especially with respect to Rossby waves. However, a sub-region at 5N is very close to the southern lateral open boundary, and as shown in Fig. 5d, the U-velocity is overestimated due to open boundary effects, which may result in unrealistic results especially with respect to the salinity advection. Meanwhile, we think the current 5 sub-regions selected, are sufficient for the purpose of our study.

As requested, we add a new figure (RF. 1, insert after Fig. 10) showing the vertical salinity profiles in ASO and NDJ, as well as the vertical displacement equivalent to a 0.6 psu salinity anomaly. A new paragraph explaining this new figure is added (line 248-257).

**Reviewer #1 Comment 2:**

*I am surprised that the amplitude of correlation coefficient for the BoB (Fig. 9) is comparable with that for the sub-regions (Fig. 11). Which region was chosen when calculating the correlation coefficient for the BoB.*

**Reply 2:**

The region of the BoB is indicated in the Fig. 1, but we admit that this figure is confusing because the BoB is marked on the map from the MPI-ESM-MR. We add a new subplot to Fig. 1 presenting the model bathymetry and our research domain (RF. 2). The exact area, which was used to calculate all BoB-related parameters is also marked in this new Fig. 1b, see line 82-85.

**Reviewer #1 Comment 3:**

*Could you clearly demonstrate the lagged period for each sub-region? It's not easy to identify this information in Fig. 11.*

**Reply 3:**

We replace Fig. 11 by the RF. 3. The new added purple line give the depth-averaged correlation from 50 m to 150 m. The lagged period for each of the sub-regions is also indicated. Of cause, we mention this in line 263-264.

**Reviewer #1 Comment 4:**
*Lines 164-173. Evidences are needed here.*

**Reply 4:**
We agree. We rephrase this part making our line of thinking more clear (line 180-190).

**Reviewer #1 Comment 5.1:**
*"The distribution of composited SSTA presents a dipole mode in the tropical Indian Ocean, which indicates the MPI-ESM-MR can reproduce IOD events." It would be better to show more evidences here.*

**Reply 5.1:**
We cite Webster et al., 1999, Deser et al., 2010. These two papers show similar pIOD SSTa pattern. We also rephrase this part (line 90-94).

**Reviewer #1 Comment 5.2:**
*The salinity of HAMSOM is obviously lower than that of the other datasets (Fig. 2). Dose this affect the results in this study?*

**Reply 5.2:**
We think this doesn't affect our results very strongly. The reason is that the coastal Kelvin waves and the associated Rossby waves appear as eigen-modes of the stratified ocean. Therefore the characteristics of these waves are mainly determined by the vertical density gradient. And since salinity gradient simulated by HAMSOM are consistent with other third-party datasets, as shown in the model validation section 2.3, we are confident that our final results are reasonable.

**Reviewer #1 Comment 5.3:**
*The salinity in the subsurface layer is less correlated with each other (Fig. 3b). Are we confident enough in the results?*

**Reply 5.3:**
We calculated the correlation metrics (RF. 4) for lines shown in Fig. 3b. From this metrics, we are confident that the seasonal change of the subsurface salinity simulated by HAMSOM is reasonable, which also provides confidence in our final results. We mention the averaged correlation coefficients (EN4: 0.63; GECCO2: 0.42; RG_Clim: 0.60; MPI-ESM-MR: 0.73; HAMSOM: 0.68) of each datasets in line 130-133.

**Reviewer #1 Comment 5.4:**
*Why is the RC-CLIM not used in Fig. 2? Why choose EN4 as the "standard" in Fig. 6? How to obtain the conclusion "Overall, the standard deviations of HAMSOM and other data sets are in close agreement" according to Fig. 6?*

**Reply 5.4:**
The RG_Clim is an Argo-based dataset, which extends from 2004 to present. The reason why we didn't use RG_Clim in Fig.2 is that the mean state of RG_Clim from 2004 to 2016 presents a different climate state compared to the climate state from 1971 to 2000, especially in light of an increasing global warming.

When plotting a Taylor diagram, the ideal situation would be to have a reference dataset based entirely on observations. Unfortunately, there is no such a dataset available. However, the EN4 dataset shown in Fig. 6 is based on an objective analysis using a global quality control, while other three datasets are model-based. Hence, the EN4 dataset is obviously the best choice, when looking for a reference state.

With regard to the last question, we think the results shown in Fig. 6b are acceptable. To make use of the advantage that HAMSOM results have a high resolution, we employed the HAMSOM grid when calculating the Taylor-diagram-related parameters. Hence, for other datasets, except for HAMSOM, a linear interpolation to the HAMSOM grid was necessary, before performing the Taylor calculation. In particular, for the subsurface salinity, the HAMSOM results are supposed to show more mesoscale features due to HAMSOM's high resolution. In contrast, for the surface salinity simulated by HAMSOM, the spatial patterns are largely determined by the low resolution atmospheric forcing fields. This explains why, compared to the EN4 data, for the sub-surface (Fig. 6b) HAMSOM results show a large standard deviation, whereas for the surface (Fig. 6a) both data sets are in close agreement. We thank the reviewer for pointing out that the explanation of Fig. 6 was not clear enough. We modify the discussion of Fig. 6 along the lines of the explanation given above (line 140-152).

**Reviewer #1 Comment 5.5:**
*Could you give more explanation of the grey lines in Fig.* 6?

**Reply 5.5:**
Yes. The grey lines indicate the axis of the centered Root-Mean-Square (RMS) difference, which is often used to quantify differences between two fields. We modify the explanation of Fig. 6 (line 150-152).

**Reviewer #2 Comment 1:**
*The paper should also start presenting the bathymetry of the regions under discussion.*

**Reply 1:**
We use the RF. 2 to replace Fig. 1. The new Figure 1b shows the model bathymetry, used in our simulation. The region of the BoB, which we defined for all BoB-related calculations, and the sponge layer are also marked in this new Figure 1b. The sponge layer helps to stabilize the model. We discuss this latter issue in line82-85.

**Reviewer #2 Comment 2:**
*If you discuss salinity anomaly in the Bay of Bengal then you should also present the typical vertical salinity (and temperature) stratification in the region. The salinity anomalies discussed seem to be very, very small compared with vertical gradients in the region. What would be the equivalent vertical displacement yielding the same salinity anomalies?*

**Reply 2:**
Thanks for this very constructive comment, which improves our manuscript significantly. We add the RF. 1 showing the vertical salinity stratification in the seasons ASO and NDJ for entire BoB and for the five sub-regions, together with a new paragraph explaining this new figure (line 248-257). Dashed lines indicate the equivalent vertical displacement yielding a 0.6 psu salinity anomaly. Obviously, the vertical salinity gradient close to the surface is relatively large, especially in the northern bay, due to the fact that heavy precipitation and strong river discharge freshens the surface water. This freshening is a local monsoon dominated seasonal feature, which mainly affects the surface and upper ocean. However, at a depth of 100 m, as this new figure indicates, a 0.6 psu salinity anomaly (suggested in Fig.10 & 12) is not negligible. In sub-regions which are significantly affected by coastal Kelvin waves and associated Rossby waves, such as SAS, EBB, and NBB, the equivalent vertical displacement of a positive 0.6 psu anomaly reaches about 20-50 m. By this means, this new figure also gives a clear indication that the wave, which we investigate in our study have a significant influence on the structure of the water column.

**Reviewer #2 Comment 3:**
*The DMI has a western and an eastern SST signal. What is the influence of the western SST signal on your results? Why didn't you only use the eastern SST anomaly for your correlation analysis? I presume you'd even get a better correlation then. Please test this.*

**Reply 3:**
We tested this as proposed; the main results are shown in RF. 5.

As one can see in the RF. 5, the correlation is not getting better, when we only use the southeastern SSTa (SETIO). This is in agreement with our physical understanding that the gradient between the two anomalies is responsible for the strength of the Kelvin wave signal.

**Reviewer #2 Comment 4:**
*It is no surprise at all to see that coastal Kelvin waves and Rossby waves propagate through the oceans. I could think of 1000s of similar studies, just focussing on different regions. To come to the point. Why makes the salinity anomalies in the Bay of Bengal significant? Do these waves play any role in the biology of the regions? Do they create any other climate feedback mechanism? If so, please try to convince the reader of the great significance of your work.*

**Reply 4:**
In our current manuscript (the pre-print version), we tried to stress the importance of our work. we already discussed the importance of subsurface salinity variability with respect to the barrier layer,

the stratification evolution, currents, eddies, the near-surface state, and air-sea energy transfer. Apparently, our current discussions need to be more convincing. We discuss the unique topographic configuration of the BoB, which is more susceptible to signals from the equator through Kelvin waves and Rossby waves than any other ocean region (line 356-359).

There is no biological module involved in our simulation, so we are not able to discuss biological processes on the basis of our HAMSOM result. However, as the IOD-related subsurface salinity anomalies and they equivalent vertical displacement suggest, these waves are expected to play an important role for the biology of the BoB. We discuss this issue in section 5 to describe the importance of our work from the biological perspective (line 382-388).

The subsurface salinity is of great importance for determining the ocean barrier layer and ocean mixed layer depth. As our results show, the BoB subsurface salinity is significantly modulated by the IOD through the Kelvin and Rossby waves. Therefore, it is expected that these wave also affect the air-sea exchange processes in the BoB, which in turn also influence the remote ocean feedback to the atmosphere. We also discuss this aspect in the section 5 (line 388-390).

**Reviewer #2 Comment 5:**
*So far, only a few pIOD events have been recorded. Does this limited number of events have any implications on the statistics presented in the paper?*

**Reply 5:**
Statistically, the size of the pIOD or nIOD sample, which is five, and the size of the climatology sample, which is thirty, are sufficient to test the hypothesis that these two populations have equal means using the Welch's t-test. Furthermore, as one can see from Figs. 10 and 12, the patterns of statistically significant anomalies are consistent with the propagation features of coastal Kelvin waves and associated Rossby waves. So, in this study, the limited number of events does not affect our conclusions. We provide more information about this issue and the Welch's t-test (line 236-239).

**Reviewer #2 Comment 6:**
*Why does HAMSOM create such strong NDJ current components of V at 80 degE and U at 5 degN? These are the at the boundaries. Is there any problem with the boundary conditions?*

**Reply 6:**
Actually, as nearly all regional models, also in HAMSOM we experienced problems at the open boundaries. Therefore we implemented a sponge layer along the southern open boundary, which was used to damp disturbances arising from inconsistencies within the prescribed boundary condition extracted from the MPI-EMS-MR. Therefore, we have excluded the sponge layer area from our analysis. Since the HAMSOM model results show a reasonable agreement with observed and already known features of the BOB on the large scale, we assume that these problems along the open boundary are not able to significantly affect our region of interest.

**Reviewer #2 Comment 7:**
*If the model domain extends to the equator, please do also show the salinity distributions extending to the equator.*

**Reply 7:**
Yes, the model domain extends to the equator. But as the new Fig. 1b (RF. 2b) indicates, we defined a sponge layer along the open lateral boundaries. Therefore, we decided to only show the results of the BoB domain indicated in Fig. 1b. We explain this issue in line 82-85.

**Response Figures**

[Figure]

RF. 1: Domain-averaged climatological vertical salinity profiles of the BoB (a) and subareas (b, c, d, e, f) during ASO (red thick line) and NDJ (blue thick line). Solid thin lines with corresponding colors indicate the salinity at 100 m depth. Dashed thin lines with corresponding colors indicate the equivalent vertical displacement, yielding a 0.6 psu salinity anomaly.

[Figure]

RF. 2: Composite of sea surface temperature anomalies during ASO of pIOD years from MPI-ESM-MR (a). The contour intervals are 0.2 C. Anomalies significant at the 95% confidence level by a two-tailed Welch's t-test are hatched with grey dots. Western tropical Indian Ocean (WTIO) and southeastern tropical Indian Ocean (SETIO), the two areas related to the dipole mode index (DMI), are marked with black boxes. The bathymetry used in the downscaling simulation (b). Our research domain, the Bay of Bengal (BoB), is marked with black borders in (a) and (b).

[Figure]

RF. 3: Lead-lag Pearson correlation coefficient between the DMI and the salinity anomaly of subareas SAS (a), EBB (b), NBB (c), WBB (d), and SBB (e), respectively, at different depths, from HAMSOM. The contour intervals are 0.1. Analysis period is from 1960 to 2005. Only significant correlation coefficients with p-value < 0.05 are shaded. Purple solid lines indicate the depth-averaged correlation over 50-150 m depth range. Purple dashed lines indicate the highest correlation and the corresponding lag.

[Figure]

RF. 4: Pearson correlation for monthly climatology of subsurface salinity shown in Fig 3b.

[Figure]

RF. 5: Time series of DMI, SSTa in SETIO, and SSTa in WTIO (a). Lead-lag Pearson correlation between the SSTa of SETIO 
[revised manuscript text omitted]

---

## Author Response (AR2)

We thank the two reviewers for the careful reading of the revised manuscript and the constructive comments.
Please find below
1. Replies for Reviewer #2
2. Response Figures (RF)

**Replies for Reviewer #2**

**Comments:**

*The major concern I have regarding this work is that the model does not include the equator. Instead the model uses a sponge layer along the equator, which was not obvious from the discussion paper.*

*My understanding is that equatorial Kelvin waves play a fundamental role in the variability of the intertropical Indian Ocean. It seems that the model does not include the equatorial forcing that leads to the creation of Wyrtki jets. It has been reported that Wyrtki jets induce coastal Kelvin waves and salinity anomalies in the Bay of Bengal. I cannot see a discussion of this important feature in the paper. Here a reference:*

*Jing WANG (2017) Observational bifurcation of Wyrtki Jets and its influence on the salinity balance in the eastern Indian Ocean, Atmospheric and Oceanic Science Letters, 10:1, 36-43, DOI: 10.1080/16742834.2017.1239506*

*I request from the authors a clarification as to whether or not the effect of Wyrtki Jets is included in their study model. If not, I insist that the authors declare the omission of equatorial processes that according to other authors play an important role in the variability of the eastern tropical Indian Ocean and the salinity budget in the Bay of Bengal. Are the missing Wyrtki Jets reflected by errors in the correlations?*

*The comparison to the IOD presented by the authors is interesting but incomplete. It seems that the model's variability is exclusively derived from the coastal wind forcing (generating coastal Kelvin waves). So, how is the wind forcing along the eastern coasts of the model domain correlated to the wind forcing along the coasts of Sumatra and Java that triggers positive IOD events? Please add more analysis of the underlying wind forcing and its relation to the IOD.*

**Replies:**

The southern boundary of our model domain is the equator. The sponge layer used in our model is specifically implemented to damp disturbances arising from inconsistencies within the prescribed boundary condition. This sponge layer scheme does not block signals from the equator (the southern boundary). As can be seen in RF. 2, the salinity anomaly in EC2 (subareas are marked in RF. 1) shows a high correlation with the prescribed boundary condition of the equator (ESB, EC1). This confirms that the equatorial processes which due to the limitation of our model domain cannot be resolved in the model itself are indirectly introduced to the BoB through the prescribed open boundary condition provided by the MPI-ESM-MR. We clarify this in lines 85-90.

Since it could be shown that MPI-ESM-MR is able to reproduce the Wyrtki Jets reasonably well (Liu et al., 2016), it can be concluded, that also the effects of Wyrtki Jets are fully accounted for in our model simulations. We thank the reviewer for pointing out the

importance of Wrytki Jets for intertropical Indian Ocean processes, which was not discussed previously in our manuscript. A related discussion is now added (lines 40-43).

Since the sponge layer does not block the signal propagation, in principle the model's variability can originate from both, the local and the remote forcing. As can be seen in RF. 2, on the interannual scale, the model's variability in EC2 is more influenced by the prescribed wind forcing in EC1 and the temperature/salinity anomaly in ESB than by the local wind. As requested, we calculated the related correlations (RF. 3). The results show that the correlation between the wind forcing along the eastern Andaman Sea coast (EC2) and the wind forcing along the coasts of Sumatra (EC1) is low (0.075 - 0.25). Thus, it can be concluded that on the interannual time scale, the model's variability is not majorly induced by the coastal Kelvin waves, which are forced by the local winds along the eastern coast of the Andaman Sea. We now discussed this issue in lines 286-290.

Regarding the last issue mentioned by Reviewer 2, her/his suspicion was correct; the wind forcing along the Sumatra coasts is highly correlated to the model's variability on the interannual scale (RF. 2d, e, f). However, the wind forcing in this area can be considered as an important component of the IOD-related equatorial Indian Ocean dynamics (Delman et al., 2016; Lu et al., 2018). As shown in RF. 3, the wind forcing in EC1 shows a high correlation with the DMI (-0.66; 0.79). As we already clarified above, the equatorial forcing can enter our model domain adequately via open boundary conditions. Consequently, the salinity anomaly we analyzed contains already all the relevant processes. Therefore, we think that adding an analysis of the underlying wind forcing at the equator as suggested by Reviewer 2, is not providing any further insight into the dynamics of the BoB, which are in the focus of our study.

Reference:

Delman, A. S., Sprintall, J., McClean, J. L., and Talley, L. D. (2016), Anomalous Java cooling at the initiation of positive Indian Ocean Dipole events, J. Geophys. Res. Oceans, 121, 5805– 5824, doi:10.1002/2016JC011635.

Liu, L., Liu, B., Han, G. et al. Assessment of the seasonal variation of simulated Wyrtki jet over the tropical Indian Ocean in CMIP5 models. Arab J Geosci 9, 676 (2016). https://doi.org/10.1007/s12517-016-2704-3

Lu, B., Ren, HL., Scaife, A.A. et al. An extreme negative Indian Ocean Dipole event in 2016: dynamics and predictability. Clim Dyn 51, 89–100 (2018). https://doi.org/10.1007/s00382-017-3908-2

**Response Figures (RF)**

[Figure]

RF. 1: Schematic diagram of subareas (a; EC1, EC2) and subsection (b; ESB).

[Figure]

RF. 2: Lead-lag Pearson correlation coefficient between the forcing and the EC2-salinity-anomalies at different depths from HAMSOM (interannual scale). Analysis period is from 1960 to 2005.

[Figure]

RF. 3: Normalized 3-month running mean of the forcing (interannual variability; a, b, c) and their correlations (d).